Cranial anatomy of Bellusaurus sui (Dinosauria: Eusauropoda) from the Middle-Late Jurassic Shishugou Formation of northwest China and a review of sauropod cranial ontogeny

Moore Andrew J. 1 djmoore@gwmail.gwu.edu
http://orcid.org/0000-0002-0126-2577 Mo Jinyou 2
http://orcid.org/0000-0003-0980-4315 Clark James M. 1
Xu Xing 3
1 Department of Biological Sciences, George Washington University , Washington, DC , USA
2 Natural History Museum of Guangxi Zhuang Autonomous Region , Nanning , China
3 Key Laboratory of Vertebrate Evolution and Human Origins, Institute of Vertebrate Paleontology and Paleoanthropology, Chinese Academy of Sciences , Beijing , China
Farke Andrew
Electronic publication date: 2018 Jun 1
Publication date: 2018
Volume: 6
Electronic Location ID: e4881
Received 2018 Mar 15; Accepted 2018 May 10
Copyright: © 2018 Moore et al.
Copyright year: 2018
Copyright holder: Moore et al.
License: This is an open access article distributed under the terms of the Creative Commons Attribution License, which permits unrestricted use, distribution, reproduction and adaptation in any medium and for any purpose provided that it is properly attributed. For attribution, the original author(s), title, publication source (PeerJ) and either DOI or URL of the article must be cited.
License URL: https://creativecommons.org/licenses/by/4.0/

Keywords: Bellusaurus, Sauropod, Ontogeny, Skull, Middle-Late Jurassic, China

Funding: National Science Foundation DGE-1246908; OISE 1515288 Jurassic Foundation, Cosmos Club, and the George Washington University National Geographic Society and National Science Foundation EAR 0922187 and EAR 0310217 National Natural Science Foundation of China 41688103, 91514302, 41120124002 Strategic Priority Research Program of the Chinese Academy of Sciences XDB18030504 Andrew Moore was supported by the National Science Foundation (DGE-1246908; OISE 1515288), Jurassic Foundation, Cosmos Club, and the George Washington University. This work was also supported by grants from the National Geographic Society and National Science Foundation (EAR 0922187 and EAR 0310217) to James Clark, and from the National Natural Science Foundation of China (No. 41688103, 91514302, 41120124002) and the Strategic Priority Research Program of the Chinese Academy of Sciences (Grant No. XDB18030504) to Xu Xing. The funders had no role in study design, data collection and analysis, decision to publish, or preparation of the manuscript.

==============================
Bellusaurus sui is an enigmatic sauropod dinosaur from the Middle-Late Jurassic Shishugou Formation of northwest China. Bellusaurus is known from a monospecific bonebed preserving elements from more than a dozen juvenile individuals, including numerous bones of the skull, providing rare insight into the cranial anatomy of juvenile sauropods. Here, we present a comprehensive description of the cranial anatomy of Bellusaurus, supplementing the holotypic cranial material with additional elements recovered from recent joint Sino-American field expeditions. Bellusaurus is diagnosed by several unique autapomorphies, including a neurovascular foramen piercing the ascending process of the maxilla at midheight, the frontal process of the nasal extending farther posteriorly onto the frontal than the prefrontal, and U-shaped medial and lateral notches in the posterior margin of the ventral process of the squamosal. Several features identified here, including a preantorbital opening in the maxilla, a stepped dorsal margin of the vomerine process of the pterygoid, and the partitioning of the dorsal midline endocranial fossae associated with the dural venous sinuses into anterior and posterior components by a transverse ridge of the parietal, are consistent with recent phylogenetic hypotheses that recover Bellusaurus as a basal macronarian or close relative of Neosauropoda. We review the current state of knowledge of sauropod cranial ontogeny, placing several aspects of the cranial anatomy of Bellusaurus in an ontogenetic context and providing explicit hypotheses of ontogenetic transformations that can be tested by future discoveries of ontogenetic variants of sauropod skulls. While scoring ontogenetically variable characters as unknown may help to alleviate the biasing effects of ontogeny on the phylogenetic position of juvenile specimens, we caution that this approach may remove phylogenetically informative character information, and argue that inference methods that are known to be less sensitive to homoplasy than equal weights parsimony (i.e., implied weights parsimony; Bayesian approaches) should also be employed.

Introduction

Sauropod dinosaurs are among the most diverse and abundant members of Mesozoic vertebrate faunas (Wilson, 2002) and reached their acme globally in the Late Jurassic (Mannion et al., 2011), with more than 20 species having been described from the North American Morrison Formation alone (Ikejiri, 2005; Harris, 2006; Tschopp, Mateus & Benson, 2015). Although an apparently comparable diversity of sauropods has been described from Middle-Late Jurassic localities in China (Dong, Zhou & Zhang, 1983; Martin-Rolland, 1999; Li et al., 2011; Xing et al., 2015b), these specimens have generally not received the scrutiny that has attended the study of North American sauropods (but see He, Li & Cai, 1988; Ouyang & Ye, 2002). Emblematic of this general dearth of taxonomic and anatomical interrogation, no fewer than 14 genera have been named from Middle-Late Jurassic strata of the Junggar and Sichuan Basins of China; despite lacking apomorphy-based diagnoses, two of these genera, Mamenchisaurus and Omeisaurus, currently comprise at least six species each, and are very likely catch-all “waste basket” taxa in need of taxonomic revision. As a result, it is unclear whether the alpha taxonomy of sauropods from Middle-Late Jurassic deposits in China mirrors that of Late Jurassic sauropods from western North America, or whether this diversity is instead an artifact of the lack of specimen-level phylogenetic and comparative analyses.

Recent studies of newly discovered specimens (Sekiya, 2011; Xing et al., 2015a, 2015b) are providing fresh insight on the anatomy and evolutionary relationships of Middle-Late Jurassic Chinese sauropods; however, untangling the taxonomy and systematics of Chinese sauropods will ultimately require redescription of historically and taxonomically important specimens in concert with discovery and description of new exemplars from across the Junggar and Sichuan Basins. To that end, we thoroughly redescribe the cranial material of the enigmatic taxon Bellusaurus sui Dong, 1990, one of four sauropod taxa recognized from the Middle-Late Jurassic Shishugou Formation of northwest China, supplementing the material described by Dong (1990) with new elements recovered in the course of joint Sino-American field expeditions of the Institute of Vertebrate Paleontology and Paleoanthropology of the Chinese Academy of Sciences and The George Washington University. We differentiate Bellusaurus from other sauropods of the Shishugou Formation and provide an apomorphy-based diagnosis and extensive morphological comparisons that indicate that Bellusaurus, despite being known solely from juvenile material, can be distinguished from other Middle-Late Jurassic Chinese sauropods. Lastly, we review the current state of knowledge of sauropod cranial ontogeny, and place several noteworthy cranial characters observed in Bellusaurus in their appropriate ontogenetic and phylogenetic contexts.

Methods

Descriptions, comparisons, and skull reconstruction

All descriptions were made directly from the holotype and referred cranial material of Bellusaurus. Comparisons with other taxa were made from direct observations of specimens or with published descriptions, illustrations, and photographs. Measurements of key dimensions of the skeletal elements were taken with Fowler electronic calipers to the nearest tenth of a millimeter and are presented in Table 1. We provide a hypothetical skull reconstruction of B. sui based on the holotypic and referred material in Fig. 1.

Table 1 Measurements (mm) of the cranial material of Bellusaurus.

Element	Measurement	Specimen	Value	
Maxilla	Length along ventral margin	IVPP V8299	46.6	
		IVPP V17768.1	98.8	
		IVPP V17768.3	83.4	
	Height of body below ventralmost point of antorbital fenestra	IVPP V17768.1	26.6	
		IVPP V17768.3	17.4	
Nasal	Anteroposterior length along midline	IVPP V17768.4	65	
	Maximum width	IVPP V17768.4	29.8	
Frontal	Greatest anteroposterior length, along interfrontal suture	IVPP V17768.5	48.7	
		IVPP V17768.6	41.2	
		IVPP V17768.7	43.6	
	Greatest transverse width, measured from interfrontal suture to lateralmost extent dorsal surface of process for postorbital	IVPP V17768.5	49.4	
		IVPP V17768.6	48.8	
		IVPP V17768.7	54.5	
	Combined breadth anteriorly of prefrontal and nasal articulations	IVPP V17768.5	32.5	
		IVPP V17768.7	41.2	
	Anteroposterior diameter of frontoparietal fenestra	IVPP V17768.5	10.7	
		IVPP V17768.6	10.5	
		IVPP V17768.7	10.9	
Parietal	Anteroposterior length along midline	IVPP V17768.5	24	
		IVPP V17768.6	28.5	
		IVPP V17768.7	24.9	
	Minimum transverse width at supratemporal fenestra	IVPP V17768.5	17	
		IVPP V17768.6	18	
		IVPP V17768.7	22	
	Transverse width of supratemporal fenestra	IVPP V17768.5	27.5	
		IVPP V17768.6	28.7	
		IVPP V17768.7	19	
	Anteroposterior width of supratemporal fenestra	IVPP V17768.5	18.6	
		IVPP V17768.6	19.6	
		IVPP V17768.7	15	
Squamosal	Maximum dorsoventral height	IVPP V17768.8	49.5	
Quadrate	Maximum dorsoventral height	IVPP V17768.9	63.8	
Parabasisphenoid	Length of basipterygoid process along medial edge	IVPP V8299	19.4	
	Proximal anteroposterior breadth of basipterygoid process	IVPP V8299	13.9	
	Proximal transverse breadth of basipterygoid process	IVPP V8299	6.4	
	Distal anteroposterior breadth of basipterygoid process	IVPP V8299	11.9	
	Distal transverse breadth of basipterygoid process	IVPP V8299	7	
Angular	Anteroposterior length	IVPP V17768.12	117.8	
	Maximum dorsoventral height	IVPP V17768.12	18.6	
Note:

Underlined values indicate minimum lengths, reflecting incomplete preservation; italicized values are estimations.

Figure 1 Reconstruction of the skull of B. sui from the Middle-Late Jurassic Shishugou Formation of Xinjiang, China.

This reconstruction is a composite based on isolated holotypic and referred material. (A) Right lateral view. (B) Dorsal view. Holotypic elements are indicated in blue and referred elements are in green.

Three-dimensional rendering of maxillae and parabasisphenoid

To produce three-dimensional reconstructions of two maxillae (IVPP V17768.1 and IVPP V17768.3) and the parabasisphenoid (IVPP 8299.2), the specimens were subjected to micro-computed tomography (CT) scanning using a Nikon XT H 225 micro-CT scanner with slice thicknesses of 62.8 μm for the maxillae and 31.4 μm for the parabasisphenoid. Data were output in raw file format and imported into Mimics v.15.0 for viewing, analysis, and visualization. Poor or ambiguous preservation of portions of the maxillary neurovasculature, especially in IVPP V17768.3, required interpolation of endocast volumes between areas where these channels were better preserved. Raw CT scan data (TIFF file stacks) are reposited on MorphoBank (http://morphobank.org/permalink/?P3122). Further details of CT scanning protocol are available from the Key Laboratory of Vertebrate Evolution and Human Origins of the Chinese Academy of Sciences.

Systematic Paleontology

SAURISCHIA Seeley, 1887

SAUROPODOMORPHA Von Huene, 1932

SAUROPODA Marsh, 1878

EUSAUROPODA Upchurch, 1995

BELLUSAURUS Dong, 1990

Type Species—Bellusaurus sui (by monotypy).

Diagnosis—As for type and only species (see below).

Occurrence and Age—Konglonggou area, Junggar Basin, Xinjiang Uyghur Autonomous Region, northwest China. The vertebrate locality producing Bellusaurus was discovered in 1954 by a Kelameili regional petroleum exploration team of the Xinjiang Petroleum Administrative Office. In 1983, the Xinjiang Paleontological Expedition of the Institute of Vertebrate Paleontology and Paleoanthropology (IVPP) of the Chinese Academy of Sciences collected hundreds of bones from an assemblage of mostly disarticulated, juvenile skeletons, from which Dong (1990) erected B. sui. In 2003, a Sino-American field expedition comprising the IVPP and The George Washington University re-opened the quarry at the area informally called Dinosaur Valley (Konglonggou), approximately 10 km northeast of the town of Huoshaoshan, and collected hundreds of additional specimens referable to Bellusaurus, including new cranial elements (Clark et al., 2006; Mo, 2013). The presence of 17 left scapulae among the material collected by the Xinjiang Paleontological Expedition and the seven additional left scapulae recovered in 2003 together indicate that at least 24 individuals are preserved in the quarry, a greater number than can be confidently inferred from any other element. No other taxa are known from the quarry.

The Bellusaurus quarry is situated at the base of the upper beds of the Shishugou Formation in a succession of tan-colored calcareous mudstones and fine-grained sandstones, near the type locality of the mammaliaform Klamelia zhaopengi Chow & Rich, 1984. The lower contact of the Shishugou Formation with the Xishanyao Formation in this area has not been located precisely, but the quarry lies 80 m stratigraphically above a coal bed in the uppermost Xishanyao Formation (D. Eberth, 2014, personal communication). The Shishugou Formation is poorly exposed in this area but the transition from lower beds (variegated redbeds and coarser-grained sandstones) into upper beds (finer-grained, tuffaceous and tan-colored mudstones and sandstones) matches a similar transition that is present in all areas where the formation is better exposed (e.g., Jiangjunmiao; Wucaiwan). At Wucaiwan—25 km to the northwest of Dinosaur Valley—the Shishugou Formation is 378 m thick (Eberth, Xu & Clark, 2010). Lithostratigraphic correlation indicates that the upper part of the formation is thicker and the lower part thinner at Dinosaur Valley than at Wucaiwan, and places the Bellusaurus quarry at the level of the lowest horizons of the upper beds of the Shishugou Formation at Wucaiwan (D. Eberth, 2014, personal communication), between two tuffs—one in the middle beds (Tuff T-1) and one in the upper beds (Tuff T-BW) (Fig. 2). Radiometric dating at Wucaiwan provides ages of 162.2 ± 0.2 and 159.7 ± 0.3 Ma for the T-1 and T-BW tuffs, respectively (Choiniere et al., 2013; Han et al., 2015), placing the Bellusaurus quarry, and much of the Shishugou Formation, within the earliest Oxfordian (early Late Jurassic) (Fig. 2).

Figure 2 Stratigraphic position of sauropods from the Middle-Late Jurassic Shishugou Formation of Xinjiang, China.

Composite section based on local sections at Jiangjunmiao, Konglonggou, and Wucaiwan. Radiometric dates are from Wucaiwan only. Copyright by David Eberth, modified with permission.

The stratigraphic positions of other Shishugou sauropods are somewhat less well-constrained (Fig. 2). Mamenchisaurus sinocanadorum was recovered from a coarse, rusty-red channel sandstone in the upper portion of Shishugou Formation, near Jiangjunmiao, and Tienshanosaurus chitaiensis Young, 1937 was excavated from an unknown stratigraphic level in roughly the same area (Russell & Zheng, 1993), while Klamelisaurus gobiensis was found in gray-brown to purple red sandy mudstones near Jiangjunmiao at the top of what was formerly called the “Wucaiwan Formation” (Zhao, 1993; Clark et al., 2006) and is now considered to constitute the lower beds of the Shishugou Formation (Clark et al., 2006).

BELLUSAURUS SUI Dong, 1990

(Figs. 3–19)

Holotype—IVPP V8299.1-7, fragmentary cranial elements including a nearly complete right otoccipital, partial parabasisphenoid, partial left maxilla, and four isolated teeth. Dong (1990) also described an isolated supraoccipital, but this element was not figured in the original description and could not be located for study.

Referred material—IVPP V8300, a composite skeleton lacking cranial elements, described by Dong (1990).

IVPP V17768.1-21, three incomplete right maxillae, a partial right nasal, an articulated left frontal and parietal, two articulated right frontals and parietals, a nearly complete right squamosal, a right quadrate, a right pterygoid, an incomplete left dentary, a nearly complete right angular, 10 isolated teeth, and postcranial elements from numerous individuals (Mo, 2013). Although none of the new elements, with the exception of an articulated sacrum, were preserved in articulation, morphological overlap among the elements described by Dong (1990) and the new material, as well as the high concentration of morphologically-consistent juvenile sauropod bones within a narrow layer of the Shishugou Formation, indicate that these elements come from a single taxon and are referable to B. sui. The presence of three right maxillae indicates that at least three individuals are represented by cranial material; there is no evidence that multiple elements came from any one individual.

Emended Diagnosis—(Cranial features only.) A non-neosauropod eusauropod near the origin of Neosauropoda (Wilson & Upchurch, 2009; Royo-Torres & Upchurch, 2012; Mo, 2013) or early-branching macronarian (Upchurch, Barrett & Dodson, 2004; Carballido & Sander, 2014) diagnosed by the following unique autapomorphies: neurovascular foramen piercing ascending process of the maxilla at midheight; frontal process of the nasal extends farther posteriorly onto the frontal than the prefrontal; U-shaped medial and lateral notches in the posterior margin of the ventral process of the squamosal, near its base; a shallow, anteromedially-facing concavity on the ventral articular process of the quadrate, extending from the medial condyle to the anteroventral edge of the pterygoid wing; and a pronounced, trough-like structure on the dorsal margin of the pterygoid at the union of the vomerine, transverse, and quadrate processes.

Differential diagnosis—Here we present differentiation of B. sui to show that this taxon is not a juvenile specimen of other named Shishugou Formation sauropods.

Klamelisaurus gobiensis has been hypothesized to be the adult form of Bellusaurus (Paul, 2010), presumably on the basis of the co-occurrence of these taxa in the Shishugou Formation and the juvenile status of all known Bellusaurus material. Klamelisaurus is from the lower beds of the Shishugou Formation and is thus stratigraphically older than Bellusaurus, falling within the late Callovian (latest Middle Jurassic) portion of the Shishugou Formation (Fig. 2). With the exception of teeth that could not be located for study, the holotype and only specimen of K. gobiensis (IVPP V9492; Zhao, 1993) does not preserve cranial elements. However, there is substantial morphological overlap between the postcranial skeletons of Klamelisaurus and Bellusaurus specimens, and Bellusaurus can be readily distinguished from Klamelisaurus in: lacking presacral neural spine bifurcation (this character may be ontogenetically variable: Woodruff & Fowler, 2012; Wedel & Taylor, 2013); having cervical vertebrae with lateral pneumatic excavations subdivided by two or more oblique accessory laminae (cervical and dorsal pneumatic excavations of the centrum are generally deeper and more extensively subdivided in Bellusaurus than they are in Klamelisaurus; this is the opposite of what would be expected if the Bellusaurus quarry were comprised of juvenile specimens of Klamelisaurus, given that pneumatic structures in sauropods progress ontogenetically from simple fossae to deeper and more extensively subdivided pneumatic recesses: Wedel, 2003; Schwarz et al., 2007; Carballido & Sander, 2014; Tschopp & Mateus, 2017); lacking cervical vertebrae with ventral surfaces that are mediolaterally concave in the anterior half; lacking posterior projections of the transverse processes of the cervical vertebrae; having a tab-like process on the prezygodiapophyseal lamina, below the pre-epipophysis, in middle-posterior cervical vertebrae; lacking sheet-like extensions of the spinoprezygapophyseal lamina of middle-posterior cervical vertebrae; having an accessory, subvertical lamina in the postzygapophyseal centrodiapophyseal fossa of posterior cervical vertebrae; having lateral pneumatic foramina of the dorsal centra with margins that are flush with the lateral surface of the centrum; having camerate dorsal centra; having dorsal vertebrae with transverse processes whose distal ends curve smoothly onto the dorsal surface of the process; having dorsolaterally directed transverse processes in middle-posterior dorsal vertebrae; lacking a posterior centroparapophyseal lamina in middle-posterior dorsal centra; having vertically oriented rod-like struts dividing the lateral pneumatic excavation of middle and posterior dorsal vertebrae; having ventral bifurcation of posterior centrodiapophyseal laminae of posterior dorsal vertebrae; having ventral bifurcation of the medial centropostzygapophyseal lamina in posterior dorsal vertebrae (likely a postcranial autapomorphy of Bellusaurus); having a divided centropostzygapophyseal lamina in middle and posterior dorsal neural arches, with the lateral branch connecting to the posterior centrodiapophyseal lamina; having middle and posterior dorsal neural spines with anteroposterior widths that are approximately constant throughout the height of the spine; lacking aliform processes that project farther laterally than the postzygapophyses in middle and posterior dorsal vertebrae (this feature may vary ontogenetically: Carballido & Sander, 2014); lacking posterior offset of neural spines in middle and posterior dorsal vertebrae; having a subtriangular lateral pneumatic foramen in the centrum of posterior dorsal vertebrae; having a smoothly rounded anterodorsal coracoid margin; having a coracoid with a distinct infraglenoid lip and notch; having a humeral head with a prominent subcircular process on the posterior surface of the proximal end; lacking a distally expanded deltopectoral crest of the humerus.

Bellusaurus differs from M. sinocanadorum—the only species of Mamenchisaurus named from the Shishugou Formation, the holotype and only specimen of which was recovered from the upper part of the Shishugou Formation at Jiangjunmiao (Russell & Zheng, 1993)—in: having a stepped, rather than essentially straight, vomerine process of the pterygoid; lacking a lingual boss near the distal edge of the teeth; having deep, subdivided lateral pneumatic fossae in the postaxial cervical centra; and lacking camellate internal pneumatic structure in cervical vertebrae.

Distinguishing Bellusaurus from T. chitaiensis—one of the first sauropods discovered in China (Young, 1937), likely from approximately the same locality as M. sinocanadorum (Russell & Zheng, 1993)—is more difficult, owing to limited morphological overlap, apparent loss of some of the holotypic material of Tienshanosaurus, and incomplete preservation of existing Tienshanosaurus material. Based on the available material (IVPP RV 37089) and the original description, Bellusaurus can be differentiated from Tienshanosaurus in: lacking presacral neural spine bifurcation; having strongly procoelous anterior caudal vertebrae; having a relatively elongate scapular blade; and having greater distal expansion of the scapular blade.

Description

General comments

We use Romerian orientational descriptors (i.e., anterior, posterior) rather than standardized terms (i.e., cranial, caudal). Given the lack of a standardized terminology for sauropod skull bones and their various processes, we follow Wilson et al. (2016) in employing morphological and orientational descriptors for cranial element processes, favoring morphological descriptors where it is convenient to do so.

Preservation of the holotype material is generally poorer than that of the referred material. The cranial elements of the holotypic and referred specimens were not discovered in articulation, as is also true for the vast majority of the postcranial material, and portions of at least 24 individuals were preserved in the quarry (Dong, 1990; Mo, 2013). All known Bellusaurus specimens are clearly juvenile. Among the five long bones of the newly referred material that have been sectioned for histological analysis, there is little or no secondary remodeling of bone tissue (Mo, 2013). Two elements display evidence indicative of periods of slowed or arrested growth—an apparent annulus in a fibula (IVPP V17768.283) and a single line of arrested growth (LAG) in an ulna (IVPP V17768.240) (Mo, 2013)—suggesting death within the first two years post-hatching. Sauropods are typically characterized by continuous deposition of highly vascularized fibrolamellar bone throughout most of ontogeny, with growth marks only appearing near adult size (Curry, 1999; Sander, 2000; Curry Rogers & Erickson, 2005; Sander et al., 2011; Cerda et al., 2017), though cyclic, zonal organization prior to deposition of an external fundamental system has been noted in the cortices of several taxa, including Patagosaurus (stratification into zones and annuli, and a LAG; Cerda et al., 2017), Apatosaurus (cyclic changes in vascularity; Curry, 1999), Janenschia (polish lines; Sander, 2000), and the island dwarf Europasaurus (LAGs; Sander et al., 2006). The presence of growth marks in some Bellusaurus specimens is thus somewhat unusual, and may reflect high seasonality of the Shishugou Formation (Eberth et al., 2001; Tütken et al., 2004), an acute period of stress (due to, e.g., disease), or taxon-specific growth patterns. In addition to skeletochronological indicators, most cranial elements exhibit the porous and striated cortical surface typical of fast-growing, juvenile bone (Varricchio, 1997; Benton et al., 2010; Marpmann et al., 2014). The preponderance of evidence for young juvenile status of all known Bellusaurus material and the roughly sub-equal size of duplicate elements in the bone bed (Andrew J. Moore, 2017, unpublished data)—including the four maxillae and three sutured frontal-parietals (Table 1)—suggest that all Bellusaurus specimens are of approximately the same age, and thus that intraspecific variation may be a greater source of differences between specimens than is ontogenetic variation.

Inferences about sutural contacts with missing elements are based in large part on topological associations observed in Camarasaurus (Madsen, McIntish & Berman, 1995; CM 11338, UMNH VP 5668, UMNH VP 5669) and other sauropods known from relatively intact skulls.

Maxilla (IVPP V8299.5, IVPP V17768.1-3; Figs. 3–7) Although none of the maxillae are preserved fully intact, the three referred elements together preserve most portions of the maxilla and provide a nearly complete picture of its morphology. The maxilla is comprised of a main body and several processes for articulation with the nasal, lacrimal, jugal, and possibly quadratojugal.

Figure 3 B. sui, holotype left maxilla (IVPP V8299.5).

(A) Lateral view. (B) Medial view. (C) Dorsal view. (D) Ventral view. (E) Anterior view. (F) Posterior view. Abbreviations: amf, anterior maxillary foramen; asp, ascending process of the maxilla; nf, narial fossa; snf, maxillary portion of the subnarial foramen. Scale bar = 3 cm.

The maxilla is the only cranial element of the holotype that overlaps with IVPP V17768. Dong (1990) described the holotype as including a portion of the right maxilla, which we recognize instead as a fragment of the left maxilla. The holotypic maxilla is largely incomplete, lacking the ascending and posterior processes and the anterior and ventral portions of the main body; though fragmentary, the occurrence of the holotypic maxilla within a monospecific bone bed, its size, and its morphological similarity with the referred maxillae all indicate that the holotypic and referred maxillae belong to a single taxon. Unfortunately, the holotype does not preserve the ascending process so the presence of the autapomorphic neurovascular foramen cannot be determined.

An ascending process projects posterodorsally and slightly laterally from the body of the maxilla; though this process is missing in IVPP V8299, its broken base also expands laterally on the maxillary body (Figs. 3A, 3D and 3E). All three of the referred maxillae preserve portions of the ascending process, though the direction of the process is crushed ventrally in IVPP V17768.1. The distal posterolateral surface of the ascending process was presumably overlapped laterally by the descending lateral process of the nasal and the dorsal portion of the lacrimal. The shaft of the ascending process is pierced by a small foramen at just over mid-height of the process (Figs. 5A, 5B and 5D). There is a shallow trough dorsal to the foramen on the lateral surface of the process and a pronounced ventral trough on the medial surface of the process, indicating passage of a neurovascular channel between the external and internal surfaces of the ascending process in a line subparallel to the trajectory of the process. This foramen and the associated short, deep trough on the medial surface of the ascending process have not been previously described for other sauropods; a similar feature may be present on the right side of Shunosaurus (ZDM 5009), though in this specimen, there is no evidence of an associated trough and the remainder of the skull exhibits taphonomic or pathological pockmarking of the bone surface that calls into question the validity of the foramen. We thus interpret the presence of a neurovascular foramen and associated trough in the ascending process of the maxilla of Bellusaurus to be an autapomorphy of the taxon.

The ascending process makes up the anterior border of the antorbital fenestra, which lacks an antorbital fossa. The last maxillary tooth is positioned just posterior to the midpoint of the antorbital fenestra; Bellusaurus thus resembles other sauropods in having the tooth row anterior to the orbit, but lacks the condition in diplodocoids and some titanosauriforms wherein the maxillary tooth row is anterior to the antorbital fenestra, though this feature may be ontogenetically variable (see Discussion).

A tapering, tongue-like premaxillary process extends anteriorly from the anteriormost portion of the palatal shelf, and is continuous posterolaterally with the ascending process via a thin sheet of bone—the “maxillary flange” of Upchurch (1998)—that bounds the narial fossa medially and posteriorly, separating the fossa from the antorbital cavity (Figs. 5 and 6). Only IVPP V17768.2 preserves an essentially complete maxillary flange. A straight, narrow groove for the dorsal maxillary process of the premaxilla extends anteriorly along much of the dorsomedial margin of the maxillary flange until it reaches the base of the premaxillary process (IVPP V17768.2), while a corresponding shallow groove for the ventral maxillary process of the premaxilla extends along the ventromedial aspect of the premaxillary process. The medial surfaces of the maxillary flange and base of the premaxillary process are smooth, and in IVPP V17768.1, this surface is very slightly concave and tapers posteriorly toward the palatal shelf. At the base of the premaxillary process, the anterior surface of the maxilla preserves a dorsally and slightly laterally directed channel that corresponds to the maxillary portion of the subnarial foramen (Figs. 3–6). The subnarial foramen is well-removed medially from the gently angled lateral margin of the narial fossa, and is partially obscured in lateral view. The narial fossa is more developed than in Shunosaurus lii (ZDM 5009), where the fossa is only weakly offset from the lateral margin of the maxilla, but shallower than that in most specimens of Camarasaurus (CM 11338, BYU 13743, UMNH VP 5907, UMNH VP 5959, UMNH VP 11393), Giraffatitan (MB.R.2180.2), and cf. Brachiosaurus (USNM 5370), in which the fossa is a sunken embayment that falls below the angular margin at the juncture of the narial fossa and the lateral surface of the maxilla. The surface of the narial fossa is smooth in IVPP V17768.1 and V17768.3, while that of IVPP V17768.2 exhibits irregular, wart-like tuberosities that extend across the narial fossa, into the subnarial foramen, and onto the anteromedial surface of the palatal shelf, above the tooth row (Fig. 5). None of the other maxillae preserve similar structures, and we interpret their presence in IVPP V17768.2 to be pathological.

Posteriorly, the main body of the maxilla flares dorsoventrally owing to the presence of a well-developed, triangular lacrimal process that occupies its posterodorsal corner and a blunt, posteroventral extension of the maxillary body that articulated with the jugal and possibly the quadratojugal (Figs. 4 and 6). The posterior margin of the maxilla is concave between these two projections. The dorsal surface of a pronounced palatal shelf meets the edge of the antorbital fenestra laterally and extends posteriorly as far as the lacrimal process, nearly reaching the posterior edge of the maxilla. The dorsal surface of the antorbital cavity is bordered anterolaterally by the ascending process, which has a large, smooth fossa on its internal surface. This large fossa of the ascending process is continuous with an anteroposteriorly elongate fossa on the dorsal aspect of the palatal shelf. Two adjacent sutural scars are present at the posteromedial extent of the palatal shelf: a nearly flat, subtriangular facet for the palatine and a larger, oval-shaped concavity immediately posterior to it, at the posterior end of the palatal shelf at a level subequal with the posterior extent of the maxillary tooth row, for the ectopterygoid; this region is damaged in IVPP V17768.3, but the facets are well-preserved in IVPP V17768.1 (Figs. 4C and 4E). In theropods, basal ornithischians, and non-neosauropod sauropodomorphs, the ectopterygoid articulates laterally with the jugal (Wilson & Sereno, 1998); by contrast, in neosauropods, the ectopterygoid articulates with the maxilla. Abrosaurus and Bellusaurus preserve morphologies that presumably reflect the anterior migration of the lateral articulation of the ectopterygoid through sauropod evolution: in Abrosaurus, only a small portion of the ectopterygoid articulates laterally with the posterior process of the maxilla, the rest of it articulating with medial face of the jugal (ZDM 5038), while in Bellusaurus, the ectopterygoid facet of the maxilla is large and pronounced, though continuity of the ectopterygoid and jugal facets (see below) suggests that the ectopterygoid may have had a small articulation with the jugal. As in Camarasaurus (e.g., CM11338, DMNH 32126, UMNH VP 5959), the palatine scar in Bellusaurus is directed medially and is less distinct than the posteromedially-facing ectopterygoid scar.

Figure 4 B. sui, right maxilla (IVPP V17768.1).

(A) Lateral view. (B) Dorsal view. (C) Medial view. (D) Anterior view. (E) Posterior view emphasizing articulations of the posterior portion of the maxilla. Abbreviations: aof, antorbital fenestra; asp, ascending process; ef, ectopterygoid facet; et, erupting tooth; jf, jugal facet; lp, lacrimal process; nf, narial fossa; pf, palatine facet; qjf, quadratojugal facet; rt, replacement tooth; snf, maxillary portion of the subnarial foramen. Scale bar = 3 cm.

Figure 5 B. sui, right maxilla (IVPP V17768.2).

(A) Lateral view. (B) Medial view. (C) Anterior view. (D) Dorsal view. Abbreviations: aof, antorbital fenestra; asp, ascending process; gdmp, groove for the dorsal maxillary process of the premaxilla; nf, narial fossa; nfo, neurovascular foramen; pmp, premaxillary process; snf, maxillary portion of the subnarial foramen; wt, wart-like tuberosities. Asterisk denotes an autapomorphy. Scale bar = 3 cm.

Figure 6 B. sui, right maxilla (IVPP V17768.3).

(A) Lateral view. (B) Medial view. (C) Dorsal view. (D) Anterior view. Abbreviations: al, alveolus; amf, anterior maxillary foramen; aof, antorbital fenestra; asp, ascending process; ef, ectopterygoid facet; lp, lacrimal process; paf, posterior alveolar foramen; paof, preantorbital foramen; pmp, premaxillary process; qjf, quadratojugal facet; snf, maxillary portion of the subnarial foramen. Scale bar = 3 cm.

The lacrimal process is immediately dorsal to the facet for the ectopterygoid. An anteroventrally directed posterior alveolar foramen for the passage of the maxillary artery and superior alveolar nerve (White, 1958) pierces the palatal shelf fossa at the anterior base of the lacrimal process; this foramen is weakly developed in IVPP V17786.1 but pronounced in IVPP V17786.3 (Fig. 6B). In the preserved maxillae, the posterior alveolar foramen is the only pronounced foramen within the palatal shelf fossa, though IVPP V17768.3 exhibits at least two small holes piercing the middle of the palatal shelf fossa, which may be incipient instances of similarly positioned, well-developed foramina present on the dorsal surface of the shelf in some specimens of Camarasaurus (e.g., UMNH VP 5959, 11393).

Abutting the posterior end of the ectopterygoid facet and continuous with the posteriormost extent of the dorsal palatal shelf is a narrow, elongate facet on the medial surface of the maxilla for articulation with the jugal (Figs. 4C and 4E). Thus, in an articulated skull, a narrow fringe of the maxilla would overlap the jugal laterally; such a simple lap joint, consisting of a laterally overlapping maxilla and medially underlapping jugal, is present in Camarasaurus (Gilmore, 1925; UMNH VP 5959), cf. Brachiosaurus sp. (USNM 5370), and Europasaurus (Marpmann et al., 2014). Notably, in Mamenchisaurus youngi, the jugal overlaps the maxilla laterally (Ouyang & Ye, 2002: p. 93), though it is possible that this articulation was somewhat more complex than a simple lap joint (see below). Partial overlap of the jugal by the maxilla can also be inferred for Turiasaurus, based on the presence of a longitudinal facet on the lateral face of the anteroventral edge of the jugal (Royo-Torres & Upchurch, 2012; CPT-1211), though in this taxon, the lateral bulging of a marked, autapomorphic boss on the external surface of the jugal produces a shallow groove that would have weakly clasped the maxilla. The internal surface of the lacrimal process is excavated at its base just anterodorsal to the narrow jugal facet, and may have accommodated an anterior projection of the jugal (Figs. 4C and 4E).

On the lateral surface of the posterior process of the maxilla, ventral to the level of the facet for the jugal, is a transversely narrow, elongate facet (19 mm in length) that extends posteroventrally to reach the posterior edge the of maxilla (Figs. 4A, 4B, 4E, 6A and 6C), indicating that an element of the zygomatic region overlapped the maxilla laterally, leaving a posteroventrally tapering external surface of the maxilla in lateral view, as in M. youngi (Ouyang & Ye, 2002: Fig. 3). Several possible identifications obtain for this facet, none of which can be asserted confidently for Bellusaurus: it could constitute an additional, lateral facet for the jugal, implying a tongue-and-groove articulation with a narrow (2 mm wide) and deep (3–8 mm) channel on the anteroventral aspect of the jugal; the narrow shelf and the lateral surface of the maxilla immediately above could be a broad articular facet for the quadratojugal; or both of these conditions may jointly apply, with the lateral facet being occupied dorsally by the jugal and ventrally by the quadratojugal. That the jugal overlaps the maxilla laterally in M. youngi suggests that this taxon could having a clasping jugal, but it is not clear whether the jugal also extended onto the medial face of the maxilla in M. youngi.

Anteroventral to the antorbital fenestra, a shallow but distinct fossa is apparent. In IVPP V17768.1, this depression lacks large foramina, exhibiting only a small subcircular nutrient foramen in its anterodorsal corner; IVPP V17768.3 exhibits the same nutrient foramen, but also bears a deep, oval-shaped neurovascular foramen that pierces the anteroventral corner of the depression (Figs. 4A, 6A, 6C and 7). That the foramen is larger than any other on the lateral surface of the maxilla, is located anteroventral to the antorbital fenestra, and communicates with a canal for maxillary neurovasculature (traceable in micro-CT scan data) suggest that it is homologous to the preantorbital fenestra (Wilson & Sereno, 1998; Martínez et al., 2016), which has been recovered as synapomorphic for Neosauropoda (Wilson & Sereno, 1998; Upchurch, Barrett & Dodson, 2004; Whitlock, 2011a) or a slightly more inclusive clade (Wilson, 2002). The structure referred to as the preantorbital fenestra by various sauropod workers exhibits substantial morphological variation (Martínez et al., 2016), ranging from a small, slit-like foramen without obvious communication with the antorbital cavity (most Camarasaurus specimens, e.g., CM 11338, CM 113; Europasaurus: Marpmann et al., 2014) to a comparatively large foramen with (Abydosaurus: DINO 17849; cf. Brachiosaurus: USNM 5370; Giraffatitan: MB.R.2180.2) or without (e.g., Jobaria: MNBH TIG 5; Bellusaurus: IVPP V17768.1, V17768.3; Dicraeosaurus: MB.R.2336.1-3) direct medial communication with the antorbital cavity to a definitive preantorbital fenestra manifesting as a broad window that is confluent medially with the antorbital cavity (e.g., cf. Diplodocus: USNM 2672; Galeamopus: Tschopp & Mateus, 2017). While the functional significance of the preantorbital opening and the full complement of morphogenetically pertinent soft-tissue structures associated with evolutionary elaboration of the foramen have not been critically evaluated across sauropods, recent work (W. R. Porter, 2015, unpublished data; Martínez et al., 2016) indicates that the preantorbital opening is vascular in origin (see Discussion).

Figure 7 Maxillary neurovasculature and replacement teeth in B. sui.

Major maxillary neurovasculature channels indicated in red; replacement teeth indicated in orange. (A) Right maxilla (IVPP V17768.1) in lateral view. (B) Right maxilla (IVPP V17768.3) in lateral view. (C) Transparent rendering of right maxilla (IVPP V17768.1) in lateral view. (D) Transparent rendering of right maxilla (IVPP V17768.3) in lateral view. Abbreviations: amf, anterior maxillary foramen; paof, preantorbital foramen. Not to scale.

In addition to the preantorbital foramen, the maxilla bears numerous other foramina surrounding the base of the ascending process and within the narial fossa (Figs. 3–7). A row of neurovascular foramina extends along the length of the lateral surface of the maxilla just above the alveolar margin, and transmitted nerve and blood vessels to the skin in life. Several foramina set within deep, elongate troughs surround the ascending process anteroventrally and extend onto its base, and generally exhibit topological consistency across the three referred maxillae. As with the other holotypic cranial material, the external surface of the holotypic maxilla is poorly preserved. However, at least three foramen-trough structures are discernible on the holotype and correspond to similarly positioned foramina on the referred maxillae: one positioned anteroventral to the ascending process and just ventral to the narial fossa, one on the anteroventral corner of the ascending process itself, and one set within the narial fossa at its lateral border, which we interpret as the anterior maxillary foramen.

Unlike the markedly sigmoid ventral margin of the maxilla in Shunosaurus (ZDM 5009), Omeisaurus maoianus (Tang et al., 2001: Fig. 8), some Camarasaurus specimens (Woodruff & Foster, 2017), brachiosaurids, and titanosaurians, Bellusaurus has a slightly convex alveolar margin in lateral view (Figs. 4A and 4C). The ventral margin of the holotypic maxilla is badly abraded and does not preserve the lateral plate that bounds the tooth row laterally in sauropods (Upchurch, Barrett & Dodson, 2004; Upchurch et al., 2007); however, this structure is nearly complete in IVPP V17768.1. Viewed ventrally, the lateral plate is essentially straight posterior to the fifth or sixth alveolus; anterior to this region, the maxilla curves gently medially. IVPP V17768.1 bears 13 alveoli, which are separated from each other by low interdental ridges that arise from the medial surface of the lateral plate. Anteriorly, the interdental ridges reach the ventral margin at an angle of approximately 70 degrees, suggesting slight procumbency of the anterior dentition. The largest teeth occur at the anterior end of the maxilla, as indicated by the gradual decrease in size of the alveoli posteriorly.

Nasal (IVPP V17786.4; Fig. 8) The nasal is a thin, plate-like bone. The dorsal surface of the nasal is flat, and the element is thinnest (∼1.5 mm) where it roofs the nasal cavity dorsally. The posterior margin of the nasal and its articulation with the frontal are missing; however, the frontal-nasal suture is preserved on the frontal (IVPP V17768.5, V17768.7) and indicates that the medial half of the nasal would have articulated with the frontal in a nearly transverse contact while the lateral half of the nasal was directed posteriorly as an acute, tab-like process that contacted the prefrontal laterally and overlapped the frontal dorsally. A triangular posterolateral process of the nasal with significant excursion posteriorly onto the frontal is likewise inferred to be present in Europasaurus (Marpmann et al., 2014: Fig. 6) and probably Jobaria (MNBH TIG 7), is less well-developed in M. youngi (Ouyang & Ye, 2002: Fig. 5A), Camarasaurus (Madsen, McIntish & Berman, 1995), Giraffatitan (MB.R.2180.22) and possibly Omeisaurus tianfuensis (He, Li & Cai, 1988: Fig. 8), and is absent in Spinophorosaurus (Knoll et al., 2012: Fig. 3C), diplodocoids (e.g., Tschopp, Mateus & Benson, 2015: Fig. 7), and somphospondylans (Martínez et al., 2016: Fig. 34). (Note that the skull reconstruction of Europasaurus depicted in Marpmann et al., 2014: Fig. 1 illustrates the prefrontal as having a large, two-pronged articulation with the frontal, rather than the medial of these two rami belonging to the nasal, as suggested by the condition in other sauropods and as depicted in Fig. 6A of Marpmann et al., 2014, though the labels for the nasal and prefrontal are switched).

Figure 8 B. sui, right nasal (IVPP V17768.4).

(A) Dorsal view. (B) Lateral view. (C) Medial view. (D) Ventral view. Abbreviations: cc, crista cranii; gnp, groove for the nasal process of the premaxilla; nc, nasal cavity; oc, orbital cavity; pmp, premaxillary process; vlp, ventrolateral process. Scale bar = 3 cm.

The nasal thickens dorsoventrally as it curves gently downward towards its ventrolateral process, which is largely missing. In other sauropods, the ventrolateral process articulates with the maxilla, lacrimal, and prefrontal. Near the medial edge of this thickened region of the nasal is a low ridge that traverses the ventral surface almost parallel to the midline internasal suture, being canted slightly anterolaterally-posteromedially. This ridge is an anterior extension of the right crista cranii (see below). The crista apparently traverses the medial edge of the ventral surface of the prefrontal (not preserved) and extends onto the nasal, dividing its ventral surface into a large, medial fossa of the nasal cavity roof and a smaller lateral fossa representing the anteriormost portion of the orbital cavity.

Anteriorly, the nasal tapers to an attenuated premaxillary process, its lateral border curving gently medially to form the dorsal rim of the bony naris. The dorsomedial surface of the premaxillary process preserves a deep, narrow groove for reception of the nasal process of the premaxilla. The groove projects medially and is deepest at its anterior end, but it shallows and its orientation becomes increasingly dorsal as it courses posteriorly, until it dissipates on the dorsal surface of the nasal.

Frontal (IVPP V17768.5-7; Figs. 9–11) Two left frontals and one right frontal are preserved. The frontal contacts its counterpart medially as well as the parietal posteriorly, the postorbital posterolaterally, the prefrontal and nasal anteriorly, and the orbitosphenoid ventrally. All three frontals are preserved in contact with their parietals; the bones are strongly sutured, but obliteration of the suture is incomplete, and it remains visible along most of its extent. The suture extends laterally from the frontoparietal fenestra, which is located on the midline of the frontal-parietal suture. As it approaches the supratemporal fenestra, the frontal-parietal suture turns anteriorly, skirting the anteromedial margin of the fenestra, which is comprised of a narrow flange of the parietal. Laterally, this flange of the parietal is stepped: a ventral projection of the parietal extends laterally, and would interlock with a dorsal and medial extension of the postorbital. Beyond its posterior contact with the parietal, the lateral wing of the frontal provides a broad, flat posterior face for reception of the postorbital (Figs. 9A, 9D, 10A, 10D, 11A and 11D). The frontal-postorbital and postorbital-parietal articulations exclude the frontal from the supratemporal fenestra, which lacks a supratemporal fossa.

Figure 9 B. sui, right frontal and parietal (IVPP V17768.5).

(A) Dorsal view. (B) Ventral view. (C) Posterior view. (D) Lateral view. (E) Medial view. Abbreviations: cc, crista cranii; dv, diploic vein impressions; dvsf, fossa associated with dural venous sinuses; fpf, frontoparietal fenestra; fps, frontal-parietal suture; lsa, laterosphenoid articular surface; naf, nasal facet; nc, nasal cavity; oc, orbital cavity; pof, postorbital facet; ppf, postparietal foramen; prf, prefrontal facet; sf, squamosal facet; sof, frontal portion of the supraorbital foramen; stf, supratemporal fenestra; tr, transverse ridge separating anterior and posterior compartments of the dural venous sinuses. Scale bar = 3 cm.

Figure 10 B. sui, left frontal and parietal (IVPP V17768.6).

(A) Dorsal view. (B) Ventral view. (C) Posterior view. (D) Lateral view. (E) Medial view. Abbreviations: cc, crista cranii; dvsf, fossa associated with dural venous sinuses; fpf, frontoparietal fenestra; fps, frontal-parietal suture; lsa, laterosphenoid articular surface; pof, postorbital facet; sf, squamosal facet; stf, supratemporal fenestra; tr, transverse ridge separating anterior and posterior compartments of the dural venous sinuses. Scale bar = 3 cm.

Figure 11 B. sui, left frontal and parietal (IVPP V17768.7).

(A) Dorsal view. (B) Ventral view. (C) Posterior view. (D) Lateral view. (E) Medial view. Abbreviations: cc, crista cranii; dvsf, fossa associated with dural venous sinuses; fpf, frontoparietal fenestra; fps, frontal-parietal suture; lsa, laterosphenoid articular surface; naf, nasal facet; pof, postorbital facet; ppf, postparietal foramen; prf, prefrontal facet; stf, supratemporal fenestra; tr, transverse ridge separating anterior and posterior compartments of the dural venous sinuses. Scale bar = 3 cm.

The frontal is dorsally concave, especially in V17768.7. At the anterolateral margin of its dorsal surface, the frontal bears two V-shaped facets (Figs. 9A and 11A). The lateral facet received the posterior process of the prefrontal, and the less acute medial facet received the posterolateral corner of the nasal. The nasal facet extends further posteriorly onto the frontal than does that of the prefrontal, a condition that we interpret as autapomorphic for Bellusaurus.

The orbital rim is deeply concave in dorsal view (Figs. 9A, 9B, 11A and 11B), as in mamenchisaurids, some flagellicaudatans (Tschopp, Mateus & Benson, 2015), Europasaurus (Marpmann et al., 2014: Fig. 6), and cf. Brachiosaurus (USNM 5730). Marpmann et al. (2014) considered the combination of a long and narrow frontal with a deep orbital rim and relatively narrow articular surface for the prefrontal and nasal to be autapomorphic for Europasaurus, and stated that the general condition in sauropods is for the anterior edge of the frontal to form an articular surface for the nasal and prefrontal that is nearly as wide as the widest section of the frontal. However, a relative reduction in transverse breadth of the anterior articular surface is known in other sauropodomorphs, and indeed, may be the plesiomorphic condition for Sauropodomorpha, with the breadth of the anterior articular surface for the nasal and prefrontal being 80% or less the width of the widest dimension of the frontal (i.e., where the posterolateral wing of the frontal extends towards the postorbital) in Lufengosaurus and Massospondylus, as well as in Jobaria, some diplodocoids, Daanosaurus, Europasaurus, and Bellusaurus (Table 2). Moreover, it is not universally true that the concavity of the orbital margin and the breadth of the anterior articular region of the frontal covary (Table 2).

Table 2 Form of orbital margin of the frontal in various sauropodomorphs.

Taxon	Specimen	Source	Transverse width across articulations for the nasal and prefrontal (X)	Greatest transverse width of the frontal in dorsal view (where it reaches the postorbital) (Y)	X/Y	Form of orbital margin in dorsal view	Clade	Stage	
Dicraeosaurus	MB.R.2336	Janensch, 1935–1936: Fig. 97	–	–	0.97	Concave	DC	LJuv	
Kaatedocus siberi	SMA 0004	Tschopp, Mateus & Benson, 2015: Fig. 7	–	–	0.93	Straight	DC	LJuv or Ad	
Camarasaurus	UMNH VP 5668	A. J. Moore, 2014, personal observation	–	–	0.92	Straight	BM	?Ad	
Camarasaurus	CM 11338	A. J. Moore, 2014, personal observation	–	–	0.90	Concave	BM	EJuv	
Qijianglong	QJGPM 1001	Xing et al., 2015b: Fig. 2	–	–	0.88	Concave	SA	Ad	
Nemegtosaurus	Z. PAL MgD-I/9	Wilson, 2005: Fig. 7	–	–	0.87	Straight	TF	?Ad	
Spinophorosaurus	GCP-CV-4229	Knoll et al., 2012: Fig. 3	–	–	0.84	Straight	SA	LJuv	
Shunosaurus	ZDM 5009	A. J. Moore, 2015, personal observation	63	75	0.84	Straight	SA	?Ad	
Tornieria	MB.R.2386	A. J. Moore, 2016, personal observation	84	100	0.84	Straight	DC	Ad	
Giraffatitan	MB.R.2180.22	A. J. Moore, 2016, personal observation	109	130	0.84	Straight	TF	Ad	
Galeamopus pabsti	SMA 0011	Tschopp, Mateus & Benson, 2015: Fig. 7	–	–	0.83	Concave	DC	LJuv	
cf. Diplodocus	USNM 2673	A. J. Moore, 2015, personal observation	–	–	0.82	Concave	DC	LJuv or Ad	
Abydosaurus	DINO 39727	A. J. Moore, 2014, personal observation	–	–	0.82	Straight	TF	LJuv or Ad	
Apatosaurus	CM 11162	Tschopp, Mateus & Benson, 2015: Fig. 7	–	–	0.80	Concave	DC	Ad	
Limaysaurus	MUCPv-205	Calvo & Salgado, 1995: Fig. 3	–	–	0.77	Straight	DC	?Ad	
Bellusaurus	IVPP V17768.7	A. J. Moore, 2014, personal observation	–	–	0.76	Concave	?BM	EJuv	
Nigersaurus	MNBH GAD512	Tschopp, Mateus & Benson, 2015: Fig. 7	–	–	0.76	Convex	DC	Ad	
Jobaria	MNBH TIG 4	A. J. Moore, 2017, personal observation	–	–	0.74	Straight	SA	Ad	
cf. Diplodocus	CM 3452	Berman & McIntish, 1978: Fig. 3	–	–	0.74	Concave	DC	?LJuv	
Lufengosaurus	IVPP V15	Barrett, Upchurch & Wang, 2005: Fig. 3	–	–	0.74	Concave	BS	?Ad	
Europasaurus	DFMMh/FV 389	Marpmann et al., 2014: Fig. 6	–	–	0.73	Concave	BM or TF	EJuv	
Europasaurus	DFMMh/FV 162	Marpmann et al., 2014: Fig. 6	–	–	0.72	Concave	BM or TF	Ejuv	
Daanosaurus	ZDM 0193 (left)	A. J. Moore, 2015, personal observation	30	42	0.71	Concave	?SA	EJuv	
Massospondylus	BP/I/4779	A. J. Moore, 2016, personal observation	22	31	0.71	Concave	BS	LJuv or Ad	
Mamenchisaurus hochuanensis	ZDM 0126	Ye, Ouyang & Fu, 2001: Plate 1.2	–	–	0.69	Concave	SA	Ad	
Bellusaurus	IVPP V17768.5	A. J. Moore, 2014, personal observation	33	49	0.67	Concave	?BM	EJuv	
Daanosaurus	ZDM 0193 (right)	A. J. Moore, 2015, personal observation	28	43	0.65	Concave	?SA	EJuv	
Massospondylus	BP/I/4934	A. J. Moore, 2016, personal observation	24	38	0.63	Concave	BS	Ad	
Europasaurus	DFMMh/FV 552	Marpmann et al., 2014: Fig. 6	–	–	0.61	Concave	BM or TF	Ad	
Notes:

“Stage” refers to the approximate ontogenetic stage of the specimen; taxa were coarsely binned into early juvenile, late juvenile or adult categories, based on gross indicators of skeletal maturity (e.g., bone texture; neurocentral fusion). Where measurements were taken in person with calipers, rather than from publications or photographs, they are provided in millimeters.

Ad, adult; BM, basal Macronaria; BS, non-sauropod Sauropodomorpha; DC, Diplodocoidea; EJuv, early juvenile; LJuv, late juvenile; SA, non-neosauropod Sauropoda; TF, Titanosauriformes.

A pronounced crista cranii divides the ventral surface of the frontal into two fossae anteriorly (Figs. 9B, 10B and 11B). The smaller anteromedial fossa housed the olfactory region of the nasal cavity and forms the dorsal margin of the anterior fenestra that transmitted the olfactory tracts of cranial nerve I into the endocranial cavity, while the larger lateral concavity roofs the orbit. In most sauropodomorphs, the frontal portion of the external rim of the orbit bears rugose ornamentation, but this margin is smooth in Bellusaurus, Qijianglong (Xing et al., 2015b: Fig. 2), Daanosaurus (ZDM 0193), Abrosaurus (ZDM 5033), Dicraeosaurus (Tschopp, Mateus & Benson, 2015), Europasaurus (Marpmann et al., 2014), Giraffatitan (MB.R.2180.22.4), and Sarmientosaurus (Martínez et al., 2016). In Bellusaurus, the orbital margin of the frontal is also noteworthy in having a defined edge at the juncture of the dorsal and ventral surfaces of the frontal, rather than a broadly rounded surface.

Posterior to the orbital cavity, the ventral surface of the frontal preserves sutural scars for the orbitosphenoid. The posterior margin of the orbital portion of the frontal forms a transversely and somewhat anteriorly oriented ridge that lies just anterior to the parietal-laterosphenoid suture and the contact between the ventromedial edge of the postorbital and the crista antotica of the laterosphenoid. A short, distinct, anterolaterally directed groove is apparent between the broad posteromedial end of the crista cranii and the medialmost contribution of the frontal to the posterior orbital surface (Fig. 9B). This groove may correspond to the dorsal margin of the supraorbital foramen, which allows passage of the supraorbital branch of the ophthalmic artery. A supraorbital foramen has generally not been described in sauropods, but has been recognized in a digital endocast of cf. Apatosaurus BYU 17096 by the presence of small, paired canals near the base of the olfactory tract that exit the skull in the dorsomedial wall of the orbit (Balanoff, Bever & Ikejiri, 2010).

A large midline foramen, historically homologized with the pineal (or parietal) foramen (Marsh, 1891; White, 1958) and present in Spinophorosaurus (Knoll et al., 2012), Europasaurus (Marpmann et al., 2014), some Camarasaurus specimens (Madsen, McIntish & Berman, 1995; Woodruff & Foster, 2017), and some flagellicaudatans (Harris, 2006; Tschopp, Mateus & Benson, 2015), is situated on the frontal-parietal suture (Figs. 9A, 9B, 9E, 10A, 10B, 10E, 11A, 11B and 11E), and here referred to as the frontoparietal fenestra. The medial edges of the frontal and parietal thin dramatically where they bound the fenestra; these edges have chipped in IVPP V17768.6 and IVPP V17778.7, but are essentially complete in IVPP V17768.5, indicating that the foramen is a genuine osseous feature of the dermal skull roof—or at least that portion of it that had ossified at death—and not an artifact of preservation. Ventrally, the posteromedial corner of each frontal makes up slightly less than a quarter of a subcircular, endocranial fossa that likely housed the pineal body and the anterior portion of the overlying system of dural venous sinuses that intervened between the neural tissue and endocranial ceiling in life (Witmer et al., 2008; Witmer & Ridgely, 2009; see below); this fossa is abruptly offset from the adjacent ventral surface of the frontal. Except where they bound the frontoparietal fenestra, the medial edges of the frontal and parietal are dorsoventrally thick, preserving pronounced sutural grooves for contact with their contralateral counterparts.

Parietal (IVPP V17768.5-7; Figs. 9–11) Two left parietals and one right parietal are nearly completely preserved. The parietal contacted its counterpart medially and is sutured to the frontal anteriorly, and also likely articulated with the postorbital anterolaterally, squamosal posterolaterally, laterosphenoid ventrolaterally, prootic ventrally, and supraocciptial and perhaps otoccipital posteroventrally. The dorsal surface of the parietal along the interparietal suture is roughly half as long anteroposteriorly as that of the frontal and forms the posterior part of the skull roof between the transversely-oriented supratemporal fenestrae. The slender anterolateral wing of the parietal comprises the anteromedial half of the supratemporal fenestra and contacted the anterodorsal process of the postorbital in a stepped sutural contact that excludes the frontal from the supratemporal fenestra. The posterior margin of the supratemporal opening is composed of the long, wing-like occipital, or posterolateral, process of the parietal, which contacted the head of the squamosal ventrolaterally. The rounded anteromedial corner of the supratemporal fenestra is gently obtuse and the posteromedial corner is acute, especially in IVPP V17768.5 & IVPP V17768.7. Although the lateral and anterolateral margins of the supratemporal fenestrae are missing, the preserved portions indicate that the length of the long axis of the fenestra was probably slightly greater than the distance separating the fenestrae. The parietal-squamosal contact is ventrally offset with respect to the parietal-postorbital suture, indicating that the temporal bar would have been shifted sufficiently ventrally to expose the supratemporal fenestra in lateral view, as in most sauropods.

At the posterior extent of the interparietal suture, the posterodorsal margin of the parietal curves laterally to meet the dorsomedial corner of the occipital face of the parietal (Figs. 9A, 9C, 9E, 11A, 11C and 11E). This smooth, curved posteromedial lip constitutes the anterolateral border of a postparietal foramen, which in life was bounded posteriorly by the supraoccipital. The postparietal foramen has previously been hypothesized to be a synapomorphy of Dicraeosauridae (Salgado & Calvo, 1992; Whitlock, 2011a) but was recently recognized as plesiomorphic for Flagellicaudata (and lost in Tornieria and Diplodocus; Tschopp, Mateus & Benson, 2015), and has a broad distribution within Sauropodomorpha, being present in Massospondylus (BP/1/4779; K. E. Chapelle, 2016, unpublished data), the basal sauropods Spinophorosaurus (Knoll et al., 2012), Nebulasaurus (Xing et al., 2015a), Qijianglong (Xing et al., 2015b), and possibly Abrosaurus (Ouyang, 1989), and the probable brachiosaurid Europasaurus (Marpmann et al., 2014).

The ventral surface of the parietal is complex. Anterolaterally, a wide, crescentic sutural contact for the laterosphenoid hugs the edge the supratemporal fenestra and abuts the frontal-orbitosphenoid contact anteromedially (Figs. 9B, 10B and 11B). The remainder of the ventral surface is dominated by endocranial fossae that correspond to the large dural venous sinuses that are common in sauropods (Witmer et al., 2008; Janensch, 1935–1936; Knoll & Schwarz-Wings, 2009; Knoll et al., 2012; Paulina Carabajal, Carballido & Currie, 2014; Martínez et al., 2016). The anteromedial corner of the parietal makes up slightly more than a quarter of the frontoparietal fenestra and fossa, and thus differs from Europasaurus, in which a similar midline skull roof aperture lacks a contribution from the frontal and is wholly bounded by the paired parietals (Marpmann et al., 2014). Two distinct pits, roughly three millimeters in diameter, dimple the dorsolateral wall of the endocranial fossa and are weakly separated by a narrow, transverse ridge (Fig. 9B); these are especially pronounced in IVPP V17768.5, and may correspond to impressions of diploic veins (Witmer et al., 2008). Posteriorly, the fossa associated with the frontoparietal fenestra is bounded by a transverse ridge that, in an articulated skull, would cross the interparietal suture between the thick medial walls of the supratemporal fenestra at the level of the posteromedial edge of the parietal-laterosphenoid suture (Figs. 9B, 9E, 10B, 10E, 11B and 11E). This ridge forms the anterior boundary of the posterior endocranial fossa that communicates with the postparietal foramen and that in life was associated with the posterior portion of the network of dural venous sinuses that overlaid the cerebrum and cerebellum in most sauropods (Witmer et al., 2008; Martínez et al., 2016). This transverse ridge is present on the endocranial ceiling of the parietal of a relatively adult Massospondylus individual (Chapelle & Choiniere, 2018: Fig. 36) but is incipiently developed or absent in more juvenile specimens (Sereno et al., 2007; K. E. Chapelle, 2016, unpublished data: Figs. 2.20A and 2.20B); it also appears to be present in Europasaurus (Marpmann et al., 2014: Fig. 7B). Moreover, the presence of this ridge is manifest in the endocasts of non-titanosaurian macronarians as the distinct division dorsally of the longitudinal dural expansion into anterior and posterior components (Janensch, 1935–1936; Knoll & Schwarz-Wings, 2009; Witmer et al., 2008; Martínez et al., 2016; confirmed through inspection of CT scans of Camarasaurus CM 11338). By contrast, this transverse ridge is absent from the endocranial ceiling (Xing et al., 2015b) and endocasts (Janensch, 1935–1936; Chatterjee & Zheng, 2002; Sereno et al., 2007; Witmer et al., 2008; Balanoff, Bever & Ikejiri, 2010; Paulina Carabajal, Carballido & Currie, 2014) of non-macronarian sauropodomorphs other than adult Massospondylus. We hypothesize that development of the transverse ridge varies both ontogenetically and phylogenetically (see Discussion).

The endocranial fossa associated with the postpartietal foramen is bordered laterally by a thick ridge of bone that would contact the prootic and the supraoccipital (Figs. 9B, 10B and 11B). This ridge also constitutes the medial boundary of an additional endocranial depression that is likely associated with the transverse (= middle cerebral) venous system, which drains into the dural sinuses (Witmer et al., 2008; Martínez et al., 2016); a bulbous expansion of the endocast ventrolateral to the posterior portion of the longitudinal dural venous sinus and corresponding to the transverse sinus system is present in Camarasaurus (Witmer et al., 2008) and is especially well-developed in Sarmientosaurus (Martínez et al., 2016), but is essentially absent in diplodocoids (Sereno et al., 2007; Witmer et al., 2008; Balanoff, Bever & Ikejiri, 2010; Paulina Carabajal, Carballido & Currie, 2014) and non-neosauropods (Chatterjee & Zheng, 2002; Knoll et al., 2012). The medial wall of this fossa is pitted with small foramina (2–4 mm in diameter; Fig. 9B), which may correspond to diploic veins (Witmer et al., 2008); in IVPP V17768.5, these manifest as a discrete posterior pit and two conjoined anterior pits, while IVPP V17768.6 bears a single foramen. The subtriangular facet for the head of the squamosal occupies the lateral portion of the ventral surface of the parietal and is adjacent to the posterior wall of the supratemporal fenestra, from which is it gently offset by a low ridge.

The wing-like occipital process of the parietal arches strongly ventrolaterally. The dorsomedial portion of the occipital surface of the process is marked by a fossa for the m. transversospinalis capitus (= m. complexus) (Tsuihiji, 2005; Button, Rayfield & Barrett, 2014). The curved dorsomedial margin of the fossa is strongly lipped, and in IVPP V17768.5 the posteriormost portion of this lip thickens to a nodular projection that is subtriangular in posterior view. Ventrolateral to the fossa for the m. transversospinalis capitus, the occipital surface of the parietal becomes gently convex as the occipital process curves to meet the squamosal anteroventrally. Medially, the occipital process bears a near-vertical suture for contact with the lateral aspect of the supraoccipital. All three parietals exhibit a stepped ventral margin to the occipital process, though this morphology is at least partly a preservational artifact, as the thin bony margin has partially chipped away in all three specimens. At its curved instep, the thin ventral margin makes up the posterodorsal roof of the endocranial fossa associated with the transverse venous system; in the absence of the supraoccipital, it is not possible to observe the presence or precise position of the external occipital foramen (= caudal middle cerebral vein foramen), which communicates with the transverse venous system and traverses the supraoccipital or the supraoccipital-parietal suture (Tschopp, Mateus & Benson, 2015) to exit onto the occipital plate (Balanoff, Bever & Ikejiri, 2010).

Squamosal (IVPP V17768.8; Fig. 12) The squamosal is tetraradiate, with an anterior process, a broad dorsomedial process, an elongate ventral process, and a shelf-like, transversely broad posterior process. This latter process does not seem to be homologous to the long, flange-like posterior process of non-sauropod sauropodomorphs, which is absent in early-branching eusauropods such as Shunosaurus (ZDM 5009) and M. youngi (ZDM 0083), and instead manifests in Bellusaurus as a result of the autapomorphic development of a pair of medial and lateral notches in the posterior margin of the ventral process near its base (see below). The squamosal is nearly complete, lacking only the tip of its ventral process and small portions of the margins of its dorsomedial and anterior processes, and articulated with the postorbital anteriorly, the parietal dorsomedially, the paroccipital process of the otoccipital posteriorly, and the quadrate and possibly the quadratojugal ventrally.

Figure 12 B. sui, right squamosal (IVPP V17768.8).

(A) Dorsal view. (B) Lateral view. (C) Posterior view. (D) Anterior view. (E) Medial view. (F) Ventral view. Abbreviations: ap, anterior process; dmp, dorsomedial process; lno, lateral notch; ltp, laterotemporal fenestra; mno, medial notch; parf, parietal facet; pof, postorbital facet; popf, paroccipital process facet; pp, posterior process; ptf, squamosal portion of the posttemporal foramen; vp, ventral process. Asterisk denotes an autapomorphy. Scale bar = 3 cm.

The anterior process of the squamosal accommodated the posterior process of the postorbital in an elongate groove that tapers in dorsoventral height posteriorly, indicating a subtriangular posterior process of the postorbital. The dorsomedial roof of the postorbital facet is largely missing, but the trajectory of the preserved portion indicates that the squamosal probably made a small contribution to the posterolateral corner of the supratemporal fenestra, as in most sauropods. The ventral margin of the anterior process is strongly arched and forms the acutely curved posterodorsal corner of the lateral temporal fenestra, the posterior border of which is provided by the sigmoid anterior margin of the ventral process of the squamosal. The lateral edge of the anterior process forms a lip-like ridge that arcs through a curve of approximately 90 degrees as it proceeds posteroventrally across the lateral surface of the ventral process, bounding a shallow lateral temporal fossa posterodorsally. This ridge gradually dissipates before reaching the posterior margin of the ventral process (Fig. 12B).

The dorsomedial process of the squamosal is a plate-like structure that projects dorsomedially to receive the occipital process of the parietal. The articular portion of the dorsomedial process is very slightly offset from the rest of the squamosal head, and there is a subtle but abrupt change in bone texture and color across the dorsal surface of the squamosal, with the medial, articular half of the squamosal head being darker and more smoothly textured than the lateral surface, which exhibits fine striae that radiate from the posterolateral corner of the head of the squamosal. A similar textural difference is apparent in Camarasaurus (UMNH VP 5594; UMNH VP 5598; UMNH VP 5665), with the lateral portion of the squamosal head, which is exposed dorsolaterally in articulated skulls, being markedly rugose.

The ventral, or quadrate, process of the squamosal is the longest of the processes. As in other sauropods, the lateral wall of the ventral process extends much farther posteriorly than does the medial wall; in Bellusaurus, the medial wall is a low lip that is most strongly developed on the proximal half. Viewed posteriorly, the lateral and medial sides of the ventral process are very slightly convex and are nearly subparallel throughout their preserved length. In lateral view, the posterior margin of the ventral process is essentially straight but for the presence of a large, posteriorly open U-shaped notch at the base of the ventral process. A shallower notch is similarly positioned on the medial aspect of the ventral process, and together these notches undercut the posterior shelf that buttresses the paroccipital process of the otoccipital to produce a posterior process (Figs. 12B, 12C and 12E). This process is reminiscent of the prong- or spur-like projection that is widespread among flagellicaudatans (Janensch, 1935–1936; Salgado & Bonaparte, 1991; Salgado & Calvo, 1992; Berman & McIntish, 1978; Tschopp & Mateus, 2013; Tschopp, Mateus & Benson, 2015; Tschopp & Mateus, 2017) and is also present in Nemegtosaurus (Wilson, 2005); however, Bellusaurus is distinct from these taxa in that the excavation beneath the posterior projection is abruptly discontinuous with the posterior margin of the squamosal, forming a U-shaped notch that has the appearance of having been “hole-punched” out of posterior margin of the ventral process and interrupting an otherwise smooth, straight edge. The presence of the U-shaped notches at the posterodorsal margins of the ventral process exposes the sulcus for the quadrate anteromedially and posterolaterally. The sulcus for the head of the quadrate is unlike the deep, cup-like cotyle that receives the quadrate head in Eoraptor (Sereno, Martínez & Alcober, 2012), Camarasaurus (UMNH VP 5594; UMNH VP 5598; UMNH VP 5665) and cf. Brachiosaurus sp. (USNM 5370) and instead manifests as a broad, shallowly concave, anteromedially canted surface that meets the posterior face of the ventral process at roughly a right angle.

Above the quadrate sulcus, the posterior process forms a steep, gently concave shelf that is distinctly offset from the dorsal surface of the quadrate head and that buttressed the paroccipital process anteroventrally. A small portion of the medial edge of the squamosal head contributes neither to the articular surface for the paroccipital process nor to the dorsomedial facet for the parietal, and is sandwiched between these two facets. This smoothly rounded edge presumably supplied the lateral boundary of the posttemporal foramen.

Quadrate (IVPP V17768.9; Fig. 13) The quadrate lacks much of its anterior, pterygoid process but is otherwise complete. Posteriorly, the quadrate bears a deep pneumatic fossa. The quadrate fossa is dorsoventrally tall, occupying 70% of the total height of the quadrate. In lateral view, the posterior margin of the quadrate is concave. The medial wall of the fossa extends farther posteriorly than does the lateral wall. Much of the lateral wall of the quadrate fossa is complete along its posterior edge, though taphonomic distortion has pushed a segment of the wall medially. Just above the ventral process for the articular and beginning just lateral to the ventrolateral corner of the quadrate fossa is a flat, dorsoventrally elongate articular surface for the quadratojugal (Fig. 13A). There is no distinct and well-preserved scar for the squamosal on the lateral surface of the quadrate, though the head of the quadrate and portions of the adjacent surfaces are generally rugose. It cannot be determined from the morphology of the quadrate or the squamosal whether the squamosal overlapped the quadratojugal ventrally, as it does in sauropodomorphs other than Flagellicaudata (Tschopp, Mateus & Benson, 2015) and perhaps Giraffatitan (Janensch, 1935–1936). On the lateral surface of the preserved portion of the pterygoid process, the quadrate takes on a smooth, polished texture that extends across most of its anterolateral surface.

Figure 13 B. sui, right quadrate (IVPP V17768.9).

(A) Lateral view. (B) Anterior view. (C) Posterior view. (D) Medial view. (E) Ventral view. Abbreviations: amc, anteromedial concavity; ptw, pterygoid wing; qf, quadrate fossa; qjf, quadratojugal facet. Asterisk denotes an autapomorphy. Scale bar = 3 cm.

Only the base of the pterygoid wing of the quadrate is preserved. On its medial face, the quadrate preserves the posterior portion of a broad, shallow fossa for the lateral side of the fan-shaped quadrate process of the pterygoid. This fossa exhibits the same polished texture as the anterolateral face of the quadrate. Unlike Camarasaurus (UMNH VP 5517; UMNH VP 5910; UMNH VP 5530; UMNH VP 6185), the medial side of the quadrate above the ventral articular process in Bellusaurus is not widely rugose and lacks an anteroposteriorly oriented ridge dividing the medial fossa into a large dorsal concavity and smaller ventral fossa (the latter feature being especially pronounced in UMNH VP 6815).

The head of the quadrate articulates with the squamosal dorsally, and is rugose and subtriangular, narrowing posteriorly to meet the medial wall of the quadrate fossa. The distal surface for the articular is finely rugose and roughly crescentic in outline, with a concave anterior edge and convex posterior edge. The articular surface is very weakly divided into two condyles; the lateral condyle does not extend as far ventrally as the medial condyle. A shallow, anteromedially-facing concavity extends dorsally from the medial condyle to the anteroventral edge of the pterygoid wing, and is separated from the remainder of the anterior surface by a low vertical ridge (Figs. 13B and 13D). The significance of this concavity is not clear, but may represent an accessory articulation for the pterygoid. We interpret the anteromedial concavity on the ventral articular process of the quadrate to be an autapomorphy of Bellusaurus.

Pterygoid (IVPP V17768.10; Fig. 14) The pterygoid is largely complete and triradiate, consisting of vomerine, transverse, and quadrate processes. Given the fragility of the vomerine process, a large piece of supporting matrix has been left intact, obscuring its medial face. The pterygoid would have articulated with its counterpart along the midline, as well as with the vomer, palatine, ectopterygoid, quadrate, and basisphenoid; these articulations are described below. As in most sauropods (Wilson, 2005), the pterygoid processes in Bellusaurus are not coplanar, and the transverse and quadrate processes project somewhat laterally relative to the plane of the vomerine process. Additionally, the base of the vomerine process is separated medially from the rest of the pterygoid by a wide, shallow trough; in dorsal view, this trough can be seen extending from the posteromedial corner of the pterygoid, just posterior to the vomerine process, to the base of the transverse process, where the trough merges with the lateral surface of the transverse process (Fig. 14C). This trough is very weakly developed in Camarasaurus (UMNH VP 5909, UMNH VP 5587) and Europasaurus (cast of DFMMh/FV 100.2 at Univ. of Chicago), but is absent in Giraffatitan (MB.R.2180.12); we provisionally consider this trough to be autapomorphic in its development in Bellusaurus. At its posterolateral extent, the trough is bounded by a narrow ridge that courses onto the dorsal margin of the quadrate process and provides the lateral wall of a narrow (4.3 mm) shelf that extends along the dorsal surface of the quadrate process. This dorsal shelf of the quadrate process is widest distally, and here it overhangs the medial fossa of the quadrate process (see below); anteriorly, the shelf turns 90 degrees to face medially, and fades into the medial surface of the quadrate process.

Figure 14 B. sui, right pterygoid (IVPP V17768.10).

(A) Lateral view. (B) Medial view. (C) Dorsal view. (D) Ventral view. (E) Anterior view. (F) Posterior view. Grayed out areas are supporting matrix. Abbreviations: bps, socket for the basipterygoid process; ds, dorsal shelf of the quadrate process; ef, ectopterygoid facet; gr, groove; ptt, pterygoid trough; qp, quadrate process; ri, ridge; tp, transverse process; vp, vomerine process. Asterisk denotes an autapomorphy. Scale bar = 3 cm.

The vomerine process is a winglike plate of bone that tapers to a point anteriorly; in life, the paired vomerine processes would have been clasped anterolaterally by the vomers. The ventral half of the vomerine process is missing, but its approximate shape is indicated by the block of supporting matrix, which preserves a partial impression of its medial face. In lateral view, the anterior process has a gently stepped dorsal margin; plesiomorphically, the dorsal margin of the vomerine process is essentially straight or gently curved (e.g., Plateosaurus: Prieto-Márquez & Norell, 2011: Fig. 19; Spinophorosaurus: Remes et al., 2009: Fig. 2; M. youngi: Ouyang & Ye, 2002: Fig. 3; M. sinocanadorum: Russell & Zheng, 1993: Fig. 1), but by Neosauropoda it becomes gently (Abydosaurus: DINO 17849) to strongly (Camarasaurus: UMNH VP 5586, UMNH VP 5909; Europasaurus: cast of DFMMh/FV 100.2 at Univ. of Chicago; Giraffatitan: MB.R.2180.12; Diplodocus: Whitlock, Wilson & Lamanna, 2010: Fig. 5; Galeamopus pabsti: Tschopp & Mateus, 2017: Fig. 13; Rapetosaurus: Curry Rogers & Forster, 2004: Fig. 26) stepped. The lateral surface of the vomerine process preserves the dorsal portion of two shallow fossae, separated anteroposteriorly by a low, broad, vertical ridge that extends ventrally from the dorsal margin of the process. Posterodorsally, above these fossae, the lateral face of the vomerine process forms a flat, subtly rugose, anterodorsally elongate surface that was described by Madsen, McIntish & Berman (1995) as marking the line of contact with the posterior edge of the palatine. At its anterodorsal extent, this surface becomes continuous with a short, longitudinal groove of unclear significance on the dorsal margin of the vomerine process (Figs. 14A, 14C and 14F); this region of the palate is poorly known in sauropods, and such a groove has not been described in other taxa preserving the pterygoid.

The transverse process is long and rod-like, as in M. sinocanadorum (Russell & Zheng, 1993: Fig. 1) but unlike most sauropodomorphs, in which it is curvilinear with a gently hooked distal end (e.g., Plateosaurus: Prieto-Márquez & Norell, 2011; Spinophorosaurus: Remes et al., 2009; Abydosaurus: DINO 17849; Camarasaurus: UMNH VP 5586; Giraffatitan: MB.R.2180.12). The process is transversely crushed distally and lacks a portion of its dorsal surface, including the sheet of bone connecting it to the vomerine process; however, this sheet seems not to have extended far anterior to the block of supporting matrix. A shallow, trough-like articular facet for the ectopterygoid extends longitudinally along the anteromedial aspect of the transverse process (Figs. 14A, 14B, 14C and 14E); the ectopterygoid articular facet is similarly positioned in Camarasaurus (UMNH VP 5586) and Giraffatitan (MB.R.2180.12), but in these taxa manifests as a deep, rugose channel.

The quadrate process lacks its posterior margin and part of the medial wall of the facet that receives the basipterygoid process of the basisphenoid. Most of the process is comprised of a dorsoventrally tall, fan-shaped plate of bone, with the ventral margin deflected strongly downward, roughly collinear with the ventral margin of the vomerine process (Figs. 14A and 14B). Such marked dorsoventral expansion of the quadrate process compares most favorably with the quadrate processes of Spinophorosaurus (Remes et al., 2009: Fig. 2D) and a juvenile diplodocid (Whitlock, Wilson & Lamanna, 2010: Fig. 5) and contrasts with the condition in most macronarians, wherein the dorsal margin of the quadrate process expands only slightly (Giraffatitan: MB.R.2180.12; Europasaurus: cast of DFMMh/FV 100.2 at Univ. of Chicago) or slopes posteroventrally (Abydosaurus: DINO 17849; Camarasaurus: UMNH VP 5586, UMNH VP 5909; Galeamopus: Tschopp & Mateus, 2017; Rapetosaurus: Curry Rogers & Forster, 2004: Fig. 26).

A shallow depression occupies the center of the lateral surface of the quadrate process. This depression and the region immediately ventral and posterior to it develop distinct, longitudinal striations, demarcating the articular region for the medial face of the pterygoid wing of the quadrate. The medial face of the quadrate process bears a low, distinct fossa posteriorly, below the narrow shelf that comprises the dorsal margin of the process. Anterior to this medial fossa, the ventral edge of the quadrate process expands anteromedially to floor the socket for the basipterygoid process, then continues dorsomedially and anteriorly to form its medial wall. The basipterygoid socket is completely intact save for a small portion of the dorsal margin of its medial wall, and the preserved portion indicates that Bellusaurus almost certainly lacked the dorsolaterally oriented hook that clasps the basipterygoid process posteriorly and is homoplastically distributed across Sauropodomorpha (Melanorosaurus: Yates, 2007, M. youngi: ZDM 0083, Turiasaurus: CPT-1211, Dicraeosaurus: Janensch, 1935–1936, Camarasaurus: CM 11338, UMNH VP 5586, 5909, 5587). The basipterygoid socket is longer anteroposteriorly than wide transversely, and extends for less than half of the preserved length of the quadrate process. Its internal surface is irregularly textured, and at its center, a single, distinct, subcircular pit, approximately four millimeters in diameter, pockmarks the socket.

Supraoccipital Dong (1990) included a supraoccipital in the holotype of Bellusaurus, but this element was not figured in the original description and could not be located for study. Dong (1990) describes the supraoccipital as similar in size and morphology to Shunosaurus, noting that it is butterfly-shaped, with a “swollen” nuchal crest on the midline and a smooth, rounded ventral notch roofing the foramen magnum.

Otoccipital (= exoccipital-opisthotic) (IVPP 8299.1; Fig. 15) In his description of the holotype of Bellusaurus, Dong (1990) described a right exoccipital, which we recognize here as the right exoccipital-opisthotic complex, or otoccipital (Sampson & Witmer, 2007; Martínez et al., 2016). This element is largely complete, but has suffered some wear and chipping on the dorsomedial, distal, and ventromedial margins of the paroccipital process and probably along its articular surfaces, which generally lack fine detail. As in most post-hatching archosaurs, no suture between the exoccipital and opisthotic can be discerned (Sampson & Witmer, 2007). The otoccipital makes up the ventrolateral portion of the occiput, and, as in most sauropods, presumably contacted the supraoccipital and prootic dorsally, the basioccipital posteroventrally, the basisphenoid anteroventrally, and the squamosal anterolaterally.

Figure 15 B. sui, holotype right otoccipital (IVPP V8299.1).

(A) Posterior view. (B) Anterior view. (C) Dorsal view. (D) Ventral view. (E) Lateral view. (F) Medial view. Abbreviations: aoc, area of the atlanto-occipital capsule; ci, crista interfenestralis; fm, otoccipital margin of the foramen magnum; mf, metotic foramen; pra, prootic articular surface; pop, paroccipital process; prof, proatlantal facet; sf, squamosal articular facet. Scale bar = 3 cm.

An anteroposteriorly elongate, foot-like ventral process comprised mostly of exoccipital (Madsen, McIntish & Berman, 1995) extends ventrally from the body of the otoccipital and bounds the foramen magnum laterally (Fig. 15), and would have articulated with the parabasisphenoid and basioccipital ventrally. Viewed medially, the outline of the foot is nearly symmetrical about the base of the ventral process. The posteriorly pointing excursion of the foot projects beyond the posterior face of the otoccipital—and thus beyond the occipital surface of the skull—where it would have contacted the basioccipital and contributed to the dorsolateral margin of the occipital condyle.

The crista interfenestralis, a tongue-like tab of bone derived from the opisthotic (Sampson & Witmer, 2007), extends ventrally from the anterior surface of the otoccipital, forming a tall, narrow channel—the cavum metoticum—that opens laterally as the metotic (or jugular) foramen and would have transmitted the jugular vein, cranial nerves IX–XI, and the distal end of the perilymphatic duct (Figs. 15E and 15F; Sampson & Witmer, 2007; Balanoff, Bever & Ikejiri, 2010). The dorsal margin of the cavum metoticum is pinched, producing a narrow groove that extends across the mediolateral width of the channel and that lies near the exoccipital-opisthotic boundary (Sampson & Witmer, 2007). Ventrally, the crista interfenestralis approaches but does not contact the anterodorsal surface of the otoccipital foot; because of this gap, the cavum metoticum would have had a small connection anteroventrally to the fenestra vestibuli (foramen ovalis). In most sauropodomorphs, the crista interfenestralis entirely separates the fenestra vestibuli from the metotic foramen, and Bellusaurus approaches this condition. However, there is broad confluence between the metotic and vestibular fenestrae in Apatosaurus (Balanoff, Bever & Ikejiri, 2010) and Nebulasaurus (Xing et al., 2015a), the latter taxon showing separate dorsal and ventral struts that incompletely divide these spaces. (Qijianglong was also described as having an incomplete crista interfenestralis (Xing et al., 2015b), but this portion of the opisthotic appears to be unpreserved (Xing et al., 2015b: Fig. 4D), prohibiting assessments of the process). It is possible that the continuity between the metotic and vestibular fenestrae in Apatosaurus, Nebulasaurus, and, to a lesser degree, Bellusaurus, results from late postnatal ossification of the crista interfenestralis (Balanoff, Bever & Ikejiri, 2010).

The ventral process of the otoccipital is pierced by a pair of foramina that likely carried posterior and anterior branches of the hypoglossal nerve (XII), as is typical for sauropodomorphs other than Amargasaurus and most titanosaurians (Wilson, Malkani & Gingerich, 2005; Paulina Carabajal, 2012; Paulina Carabajal, Carballido & Currie, 2014), which bear a single hypoglossal nerve foramen, and Camarasaurus, specimens of which may have one or two foramina for the hypoglossal nerve (Witmer et al., 2008; Paulina Carabajal, 2012; E. Tschopp, 2018, personal communication). Posteroventral to the metotic foramen, the lateral face of the ventral process bears a conspicuous, dorsoventrally tall, oval-shaped hypoglossal foramen and a small, elongate hypoglossal foramen, formerly infilled by matrix and seemingly unrecognized by Dong (1990). The medial face likewise bears two oblong foramina—a smaller posteroventral opening and a larger foramen at the base of the ventral process—though these are more vertically arrayed than the lateral foramina, which are positioned in an essentially horizontal plane. We note that our interpretation of the foramina of the otoccipital differs from that of Dong (1990), who seems to have recognized only two foramina: an anterior foramen, which we interpret as the metotic foramen and that he hypothesized to accommodate cranial nerve X, and a posterior foramen, which we interpret as the posteriormost hypoglossal foramen and that he ascribed to cranial nerves XI and XII.

The paroccipital process extends laterally and slightly posteriorly from the body of the otoccipital. As in most sauropods, the paroccipital process is anteroposteriorly compressed, forming an elongate oval in cross-section. The ventral margin of the paroccipital process is straight, and forms an angle of approximately 114 degrees with the lateral margin of the ventral process. Although the paroccipital process is incomplete distally, the pronounced flaring of its dorsal margin near the distal extent of the preserved portion indicates that the process likely did not extend much farther laterally, as this dorsal flaring occurs near the terminus of the paroccipital process in other sauropods.

The medial half of the anterior face of the otoccipital bears a suture for contact with the prootic. Though poorly preserved generally, this suture clearly encompasses the anterior face of the crista interfenestralis and extends dorsolaterally onto the anterior surface of the paroccipital process, ending abruptly just beyond the midpoint of the process. The anteroventral surface of the paroccipital process is gently concave and preserves fine striae that we interpret as marking the contact with the posterior process of the squamosal, which buttresses the paroccipital process anteroventrally. If correct, this buttressing configuration would have prevented contact between the quadrate and paroccipital process, contra Dong (1990). Shifting the position of the paroccipital process ventrally, such that it covers the dorsal portion of the quadrate articular fossa of the squamosal, would leave the articular facet of the posterior process of the squamosal unoccupied, and manually articulating these isolated elements indicates that such a configuration would produce a large, unrealistic gap between the parietal dorsally and the paroccipital process ventrally. The smooth medial margin of the squamosal suggests that Bellusaurus would have had a well-defined posttemporal fenestra. However, the lack of the supraoccipital and slight wear on the dorsomedial margin of the paroccipital process make it unclear whether the parietal would participate in the margin of the posttemporal fenestra, as in most sauropods, or be excluded from participation in the fenestra by a spur of the supraoccipital, as in M. youngi (ZDM 0083, Ouyang & Ye, 2002) or by an extension of the exoccipital (the “posttemporal process” of Harris, 2006), as in flagellicaudatan diplodocoids (Tschopp, Mateus & Benson, 2015).

A shallow fossa embays the posterior face of the otoccipital ventromedially, at the base of the paroccipital process (Fig. 15A). Among sauropods, this fossa is especially pronounced in Apatosaurus (BYU 17096) and Amargasaurus (Paulina Carabajal, Carballido & Currie, 2014), and has been reconstructed as accommodating m. iliocostalis capitis in some sauropods (Button, Rayfield & Barrett, 2014; Paulina Carabajal, Carballido & Currie, 2014) and theropods (Snively & Russell, 2007); however, insertion of m. iliocostalis capitis on the paroccipital process appears to be autapomorphic for Crocodylia (Tsuihiji, 2005, 2010), and the fossa at the ventromedial base of the paroccipital process may be more parsimoniously interpreted as being encompassed by the atlanto-occipital capsule, as in extant crocodylians and avians (Tsuihiji, 2010).

In Bellusaurus, a distinct prominence, which we interpret as a proatlantal facet, lies dorsomedial to the atlanto-occipital capsule fossa, bordering the foramen magnum laterally and interrupted dorsomedially at the supraoccipital-otoccipital suture (Figs. 15A, 15C and 15F). The ventral margin of the proatlantal facet is more sharply offset from the occipital surface than is its dorsolateral edge, which grades smoothly onto the paroccipital process. Proatlantal facets are present in some non-sauropod sauropodomorphs (e.g., Lufengosaurus: Barrett, Upchurch & Wang, 2005; Massospondylus: BP/1/4934). Wilson, Malkani & Gingerich (2005) described an isolated braincase from the Pab Formation as having pronounced proatlantal facets, and noted that proatlantal facets are uncommon in sauropods; however, whereas well-developed proatlantal facets are generally absent in mamenchisaurids (M. youngi: ZDM 0083) and non-titanosaurian macronarians (Camarasaurus: CM 11338, USNM 13786, UMNH VP 5668, UMNH VP 5669; Giraffatitan: MB.R.2180.22; Europasaurus: cast of DFMMh/FV 581.1 at Univ. of Chicago; Sarmientosaurus: Martínez et al., 2016), these structures are otherwise broadly distributed in Sauropoda, being present in the early-branching sauropods Spinophorosaurus (Knoll et al., 2012) and Shunosaurus (ZDM 5009), the indeterminate sauropod Daanosaurus (ZDM 0193), the turiasaurians Losillasaurus (Lo-26 a&b), Turiasaurus (CPT-1211, contra Royo-Torres & Upchurch, 2012), Mierasaurus (Royo-Torres et al., 2017), and Moabosaurus (Britt et al., 2017), flagellicaudatans (e.g., Apatosaurus: Balanoff, Bever & Ikejiri, 2010; Amargasaurus: Paulina Carabajal, Carballido & Currie, 2014: Fig. 1C; Kaatedocus: Tschopp & Mateus, 2013; Galeamopus: Tschopp & Mateus, 2017), and some titanosaurians (e.g., Quaesitosaurus: Wilson, 2005; Isisaurus: cast of GSP-UM7000 at Univ. of Chicago).

Parabasisphenoid (IVPP V8299.2; Fig. 16) The holotypic basisphenoid described by Dong (1990) is relatively complete. The basisphenoid floors the endocranium, and would have contacted the basioccipital posteriorly, the prootic and laterosphenoid dorsally, the parasphenoid anteriorly, and the otoccipital posterodorsally (Madsen, McIntish & Berman, 1995; Balanoff, Bever & Ikejiri, 2010; Marpmann et al., 2014). The parasphenoid rostrum is broken anteriorly, and none of the process is preserved; this breakage did not occur along a suture and no basisphenoid-parasphenoid sutural scar is apparent in the preserved bone, supporting previous inferences (Madsen, McIntish & Berman, 1995) that the parasphenoid fuses to the basisphenoid and their suture is obliterated early in sauropod ontogeny. We thus follow other authors (Balanoff, Bever & Ikejiri, 2010; Marpmann et al., 2014) and refer to the whole element as the parabasisphenoid.

Figure 16 B. sui, holotype parabasisphenoid (IVPP V8299.2).

(A) Dorsal view. (B) Anterior view. (C) Right lateral view. (D) Left lateral view. (E) Posterior view. (F) Ventral view. Abbreviations: bpp, basipterygoid process; bt, basisphenoid portion of basal tubera; cv, crista ventrolateralis; dos, dorsum sellae; fo, foramen; icf, internal carotid foramen; st, sella turcica; su, sagittal sulcus separating basal tubera. Scale bar = 3 cm.

The parabasisphenoid preserves several sutural surfaces, but their precise boundaries are generally difficult to make out. A vertically oriented and transversely concave posterior suture marks an extensive zone of contact for the basioccipital. A narrow, sagittal sulcus separates the ventral half of the parabasisphenoid portion of the basal tubera, but the tubera—or at least their parabasisphenoid component—are nonetheless closely appressed throughout their length, differing from the widely divergent basal tubera of the mamenchisaurid Qijianglong (Xing et al., 2015b), the turiasaurians Turiasaurus (CPT-1211), Losillasaurus (Lo-26 a&b), and Moabosaurus (Britt et al., 2017), and the early-branching macronarians Camarasaurus (USNM 13786, UMNH VP 5668, UMNH VP 5668), Europasaurus (cast of DFMMh/FV 581.1 at Univ. of Chicago) and Giraffatitan (MB.R.2180.22.2). Micro-CT scanning confirms that this sulcus lacks connection with the sella turcica, which housed the pituitary body in life, and thus that the craniopharyngeal canal—a remnant of the embryonic Rathke’s pouch (Baumel & Witmer, 1993)—was sealed off from the sella turcica early in ontogeny, unlike in Apatosaurus (Balanoff, Bever & Ikejiri, 2010), Giraffatitan (Knoll & Schwarz-Wings, 2009), and the Uzbekistan titanosaurian (Sues et al., 2015), in which the craniopharyngeal canal remains patent in more adult specimens.

Articular surfaces for the prootic and the laterosphenoid occupy most of the dorsal surface of the parabasisphenoid, extending from the edge of the basioccipital suture anteriorly onto the dorsum sellae, which bounds the sella turcica posterodorsally (Figs. 16A–16E). The dorsum sellae is incompletely preserved, and is missing much of its right half. A narrow, shallow sagittal trench, corresponding to the anterior extension of the midline trough that sometimes embays the dorsal aspect of the basioccipital in sauropods, separates the left and right prootic sutural scars and tapers anteriorly as it approaches the dorsum sellae.

The interior of the sella turcica is exposed anteriorly due to the incompleteness of the element. Several neurovascular structures may be associated with the sella turcica, and in Bellusaurus, the most clearly preserved of these are the internal carotid foramina. The carotid foramina pierce the lateral walls of the parabasisphenoid at a height subequal to the apex of the notch between the basal tubera and enter the sella turcica posteroventrally (Figs. 16C and 16D). Externally, each carotid foramen is set within a shallow, subvertically oriented groove on the lateral surface of the parabasisphenoid that is bordered posteriorly by the basal tuber and anteriorly by a low, dorsoventrally oriented ridge corresponding to the ventralmost extent of the crista prootica (= crista otosphenoidalis). The internal carotid groove arises posteroventrally, below the basal tubera, then turns slightly dorsally toward the internal carotid foramen. At this turn, the groove is bounded medially by a fine, rugose ridge extending from the anterolateral edge of each basal tuber onto the base of the basipterygoid process (Figs. 16C and 16D); this ridge may correspond to the crista ventrolateralis, which is rarely described in sauropods (but see Balanoff, Bever & Ikejiri, 2010) but is more commonly described in theropods (Sampson & Witmer, 2007).

Though the dorsum sellae is incompletely preserved, part of its median, ventral portion appears to be comprised of smooth, finished bone. If correct, this indicates the presence of a sagittal foramen piercing the dorsum sellae (Figs. 16B and 16E). A similarly positioned median canal is present in other sauropodomorphs, including Plateosaurus (Galton, 1985: Fig. 4E), Chebsaurus (Läng & Mahammed, 2010), Spinophorosaurus (Knoll et al., 2012), M. youngi (ZDM 0083), Apatosaurus (Balanoff, Bever & Ikejiri, 2010), Europasaurus (cast of DFMMh/FV 581.1 at Univ. of Chicago; Marpmann et al., 2014), Giraffatitan (Knoll & Schwarz-Wings, 2009), Bonatitan (Paulina Carabajal, 2012), and the Uzbekistan titanosaurian (Sues et al., 2015), and has been interpreted as accommodating either the basilar artery (Galton, 1985; Knoll et al., 2012; Marpmann et al., 2014)—the anastomosis of paired, posterior branches of the internal carotid arteries that runs posteriorly beneath the brain (Knoll et al., 2012)—or a part of the ventral longitudinal sinus (Sues et al., 2015; Martínez et al., 2016). In Bellusaurus, the passage of the paired abducens nerves (VI) is not apparent under gross inspection or micro-CT imaging; we suspect this is due to the generally poor preservation of the dorsum sellae and anterolateral aspects of the parabasisphenoid and expect that, as in non-titanosaurian sauropods, the abducens nerve traversed the sella turcica rather than passing lateral to it (Paulina Carabajal, 2012).

Anterior to the basal tubera, the basipterygoid processes project ventrolaterally and perhaps somewhat posteriorly, though without the rest of the braincase the extent of any posterior deflection is unclear. The basipterygoid processes diverge from each other at an angle of approximately 56 degrees and are relatively short, being less than twice as long as the longest diameter of the process at its base. In cross-section, the shafts of the basipterygoid processes are oval to weakly triangular in shape. The terminus of each basipterygoid process is subtriangular and rugose, for articulation with the basipterygoid socket of the pterygoid. The ventral surface of the basisphenoid preserves a broad fossa that extends between the basipterygoid processes to the anterior edge of the basal tubera. As in Apatosaurus (BYU 17096) and Galeamopus sp. (USNM 2673) but unlike M. youngi (ZDM 0083), Losillasaurus (Lo-26 a&b), Camarasaurus (UMNH VP 5669), Europasaurus (cast of DFMMh/FV 581.1 at Univ. of Chicago), and Giraffatitan (MB.R.2180.22.4), no transverse ridge connects the posterior edges of the two basipterygoid processes, and thus the ventral fossa is continuous and uninterrupted anteroposteriorly.

Dentary (IVPP V17786.11; Fig. 17) A segment of the posterior half of the left dentary is preserved. Transverse crushing of the anterior part of the element has compressed the dentigerous shelf and preserved teeth mediolaterally. The ventral margin of the dentary is straight for much its length, arcing ventrally and curving slightly medially in the anterior third of the element. At least eleven nutrient foramina pierce the lateral surface of the dentary on its anterior half, roughly corresponding to the dentigerous portion. Some of these foramina are relatively large (the largest measures 6 × 3 mm) and several are accompanied by a posteriorly shallowing trough. Like the maxilla, the dentary bears a lateral plate that borders the teeth labially and is marked by low interdental ridges on its medial face; the interdental ridges are generally less pronounced than those on the maxilla, and become difficult to discern posteriorly.

Figure 17 B. sui, left dentary (IVPP V17768.11).

(A) Lateral view. (B) Medial view. (C) Ventral view. (D) Dorsal view. (E) Close up of dentary teeth. Abbreviations: dr, dorsal ramus; Mg, Meckelian groove; vr, ventral ramus. Scale bar = 3 cm (A–D) and 1 cm (E).

The preserved portion of the dentary preserves at least seven alveoli, and possibly an eighth, though transverse crushing of the element makes this assessment difficult. In Camarasaurus, the curvature of the dentary toward the symphysis begins at the level of its minimum vertical depth and the seventh alveolus (Madsen, McIntish & Berman, 1995); in Tazoudasaurus, this curvature begins at the level of the fifth alveolus (Allain & Aquesbi, 2008), at approximately the eight alveolus in Jobaria (MNBH TIG 5), and at the sixth alveolus in Astrodon (USMN 5669). In Bellusaurus, this curvature begins at the level of the third preserved alveolus; interpolating from patterns exhibited in other sauropods, each dentary would likely have housed 9–13 alveoli, far fewer than the 18 or more dentary teeth typical of most non-neosauropods, including Massospondylus (BP/1/4934), Tazoudasaurus (Allain & Aquesbi, 2008), Mamenchisaurus (Russell & Zheng, 1993; Zhang, Li & Zeng, 1998; Ouyang & Ye, 2002), Omeisaurus maoianus (Tang et al., 2001), and perhaps Jobaria (MNBH TIG 5).

The medial surface of the dentary is marked by a pronounced, anteriorly tapering embayment that is bounded by two rami: a dorsal ramus comprised of the dentigerous shelf (which, along with its non-dentigerous posterior portion, constitutes the “medial process” of Madsen, McIntish & Berman, 1995) and a ventral ramus comprised of a medially offset strip of the dentary (the “lateral process” of Madsen, McIntish & Berman, 1995). The Meckelian groove becomes a distinct, ventromedially oriented channel at the anterior union of the two rami and continues anteriorly to the broken distal edge of the dentary (Fig. 17B).

An anteriorly expanded dentary ramus exceeding the minimum depth of the ramus by approximately 150% has previously been recovered as a synapomorphy of Eusauropoda (Upchurch, 1998; Wilson & Sereno, 1998; Wilson, 2002). This character cannot be confidently assessed in the incompletely preserved Bellusaurus dentary, but measurements of the dorsoventral heights of the element posteriorly (31 mm) and anteriorly (28 mm) indicate that the dentary decreases slightly in depth towards the mandibular symphysis, at least briefly. A decrease in the dorsoventral height of the dentary anterior to the coronoid and a subsequent increase in height toward the mandibular symphysis is typical for eusauropods, and occurs in taxa such as Abrosaurus (ZDM 5033), Jobaria (MNBH TIG 5), Camarasaurus (UMNH VP 11394), and Abydosaurus (DINO 17848).

Angular (IVPP V17768.12; Fig. 18) The angular is a simple, elongate, platelike bone, composed of a mediolaterally expanded ventral body and a dorsolateral laminar flange. The anterior half of the lateral surface of the angular has been abraded and mediolaterally crushed, but the remainder of the angular, including the exceedingly thin dorsal margin of its dorsolateral lamina, is mostly intact. Anteriorly, the angular would have been sandwiched laterally by the dentary and medially by the splenial (Madsen, McIntish & Berman, 1995); a facet for the splenial is apparent on the anterior half of the ventral body of the angular, indicating contact with the splenial ventrally and medially. Immediately posterior to the articulation of the splenial, the angular bears a facet for the prearticular, which extends principally along the ventral aspect of posterior half of the angular (Fig. 18B).

Figure 18 B. sui, right angular (IVPP V17768.12).

(A) Lateral view. (B) Medial view. (C) Dorsal view. (D) Ventral view. Abbreviations: add, floor of the adductor chamber; spf, splenial facet; paa, prearticular facet. Scale bar = 3 cm.

The transverse width of the ventral body is greatest posteriorly, forming a pronounced horizontal shelf that would floor the adductor fossa. The shelf narrows and rotates medially as the breadth of the ventral body tapers from a posterior maximum of 7 to 3 mm at its distal tip. The angular is laterally convex along its length, and its ventral margin is gently sigmoid in lateral view, unlike the more markedly sinuous ventral margin of Camarasaurus (UMNH VP 6814). The dorsoventral height of the angular increases for the posterior third of its length, then decreases gently to the blunt, distal tip of the bone. Where the dorsal lamina is tallest, it would overlap the lateral surface of the surangular (Madsen, McIntish & Berman, 1995).

Dentition (IVPP V8299.3, V8299.4, V8299.6; IVPP V17768.12-21; Figs. 4, 7, 17, 19–21; Table 3) Numerous teeth are preserved in situ across the four maxillae and at least three are preserved within the dentary (IVPP V17786.11). Dong (1990) also included six isolated teeth in the Bellusaurus holotype (IVPP 8299), of which four have been located, and the referred material (IVPP 17768) includes an additional ten isolated teeth. As described below, in situ maxillary and dentary teeth exhibit distinct morphologies, and in all cases, the morphology of the isolated teeth is consistent only with that of the in situ maxillary teeth.

Figure 19 B. sui, holotype isolated teeth (IVPP V8299.3, V8299.4, V8299.6, V8299.7).

(A, E, I, M) Lingual views. (B, F, J, N) Labial views. (C, G, K, O) Mesial views. (D, H, L, P) Distal views. (A–D) IVPP V8299.3. (E–H) IVPP V8299.4. (I–L) IVPP V8299.6. (M–P) IVPP V8299.7. Scale bar = 3cm.

Figure 20 B. sui, referred isolated teeth (IVPP V17768.12-21).

(A, E, J, M, Q, U, Y, CC, GG, KK) Lingual views. (B, F, J, N, R, V, Z, DD, HH, LL) Labial views. (C, G, K, O, S, W, AA, EE, II, MM) Mesial views. (D, H, L, P, T, X, BB, FF, JJ, NN) Distal views. (A–D) IVPP V17768.12. (E–H) IVPP V17768.13. (I–L) IVPP V17768.14. (M–P) IVPP V17768.15. (Q–T) IVPP V17768.16. (U–X) IVPP V17768.17. (Y–BB) IVPP V17768.18. (CC–FF) IVPP V17768.19. (GG–JJ) IVPP V17768.20. (KK–NN) IVPP V17768.21. Scale bar = 3cm.

Figure 21 B. sui, replacement teeth of the right maxilla (IVPP V17768.1).

(A) Medial view of entire maxilla. (B) Medial view of the in situ maxillary teeth only. Orange indicates the first generation of replacement teeth, blue indicates the second generation of replacement teeth, and red denotes neurovasculature. Numbers denote alveolus position. Abbreviations: paf, posterior alveolar foramen.

Table 3 Measurements (mm) of the teeth of B. sui.

Specimen	Total apicobasal height of crown + root	Total apicobasal height of crown	Maximum mesiodistal width of crown	Mesiodistal width of crown at its base	Slenderness index (crown height/max. breadth)	
IVPP V8299.3	21.1	18.8	8.5	–	–	
IVPP V8299.4	17.9	17.9	8.8	–	–	
IVPP V8299.6	21.8	21.8	9.5	–	–	
IVPP V8299.7	18.9	18.9	8.1	–	–	
IVPP V17768.12	20.5	16.6	7.8	5.1	2.1	
IVPP V17768.13	38.0	20.9	8.2	6.3	2.5	
IVPP V17768.14	31.7	19.7	8.6	5.6	2.3	
IVPP V17768.15	13.7	13.7	–	–	–	
IVPP V17768.16	12.7	11.4	5.9	4.7	1.9	
IVPP V17768.17	32.0	16.6	7.5	5.7	–	
IVPP V17768.18	14.4	11.1	6.5	5.4	1.7	
IVPP V17768.19	21.3	8.2	6.1	4.6	1.3	
IVPP V17768.20	19.8	17.5	7.5	–	–	
IVPP V17768.21	32.1	13.0	6.4	5.2	2.0	
Note:

Underlined values indicate minimum lengths, reflecting incomplete preservation; italicized values are estimations.

None of the teeth preserved in situ show evidence of wear, and only the first and fourth alveoli of IVPP V17768.1 preserve teeth that have erupted beyond the margin of the medial plate (Figs. 4C and 21). Additional replacement teeth are visible through the row of replacement foramina that line the alveolar margin in IVPP V17768.1. Micro-CT scans reveal that, including an erupting tooth, each alveolus houses up to two replacement teeth (Figs. 7 and 21). The second generation of replacement teeth is only preserved in situ in alveoli 1–5 and 7; more posterior alveoli only house a single tooth, and alveolus 10 does not preserve any teeth. Replacement teeth are imbricated, with more anterior teeth generally overlapping the next most posterior tooth mesiolabially. The second generation replacement tooth in a given alveolus is positioned somewhat mesiolingual to the first generation tooth in that alveolus. It is possible that the lack of a second generation of replacement teeth in more posterior alveoli reflects differential rates of tooth replacement along the tooth row, as has been noted for some other sauropods (Sereno & Wilson, 2005; Wiersma & Sander, 2016). Ontogenetic influences on tooth replacement rates and number are not known for sauropods.

The crowns of the maxillary and isolated teeth have mesiodistally convex labial surfaces and concave or nearly flat lingual surfaces, and the base of the crown has a D-shaped cross-section (Figs. 4C, 4D and 19–21). The crowns curve lingually and are gently recurved distally, especially in isolated teeth with smaller crowns, presumably originating from more posterior positions in the tooth row. At approximately midheight, the crowns of the maxillary and isolated teeth expand slightly mesiodistally, then taper to an apical point. The increase in crown breadth is much less dramatic than in the distinctly spatulate teeth of taxa such as Chebsaurus (Läng & Mahammed, 2010: Fig. 6), Tazoudasaurus (Allain & Aquesbi, 2008: Fig. 7), and turiasaurians (Royo-Torres & Upchurch, 2012: Fig. 7; Mateus, Mannion & Upchurch, 2014: Fig. 3), and some teeth (e.g., IVPP V17768.13, IVPP V17768.17, IVPP V17768.20) resemble the more parallel-sided condition of M. sinocanadorum (IVPP V10603), M. youngi (Ouyang & Ye, 2002: Figs. 8–11), Euhelopus (PMU 24705/1a-b), and brachiosaurids (e.g., Giraffatitan MB.R. 2180.20.12; Mannion, Allain & Moine, 2017: Fig. 9). A low ridge extends apicobasally along the midline of the lingual surface of the crown, from the mid-crown mesiodistal expansion to the apex of the tooth, producing shallow troughs on either side. The teeth of non-diplodocoid and non-titanosaurian sauropods also exhibit apicobasal labial grooves near their mesial and distal margins (Upchurch, 1995, 1998; Wilson & Sereno, 1998; Mannion et al., 2013; Mannion, Allain & Moine, 2017)—a feature identified as a synapomorphy of eusauropods and their closest outgroups (Wilson & Sereno, 1998; Carballido & Pol, 2010; Carballido et al., 2017)—but these are very weakly developed in Bellusaurus, manifesting as flat or shallowly concave surfaces. Tooth size decreases distally along the tooth row, and smaller teeth bear a shorter crown and are more distinctly recurved.

The maxillary teeth were closely spaced in the tooth row; this is clear not only from inspection of the preserved maxillae and their in situ teeth (IVPP V 8299; IVPP V17768.1), but is also suggested by the nearly-flat facet on the distolingual aspect of the isolated teeth that extends between the base of the crown and the mid-crown mesiodistal expansion (Figs. 19 and 20), suggesting that the distal edge of the tooth slightly overlapped the mesiolabial aspect of the succeeding tooth. This relationship cannot be confirmed in the preserved maxillae, which lack well-exposed, sequentially preserved teeth, but the presence of such a facet suggests that Bellusaurus had imbricated teeth.

Denticles are absent on both the mesial and distal margins of the maxillary and isolated teeth (contra Dong, 1990). As in other sauropods, the enamel on the maxillary teeth is wrinkled throughout the crown, most closely resembling morphotype I of Holwerda, Pol & Rauhut (2015): the enamel is arranged into coarse, longitudinal ridges at the base of the crown, below the mesiodistal expansion, with crests and grooves becoming shorter and mesially and distally deflected closer to the apex of the crown. In some isolated maxillary teeth but in none of the in situ teeth, the finely wrinkled texture of the enamel has been partially worn away from the mesial and distal margins of the upper crown (IVPP V8299.3; IVPP V17768.16-21), producing a polished finish apically and several forms of wear pattern: paired mesial and distal V-shaped facets (IVPP V8299.3, IVPP V17768.17, IVPP V17768.20), a single, mesial V-shaped facet (IVPP V17768.18, IVPP V17768.19), and a single apical wear facet (IVPP V17768.21), though this may reflect the early stages of a low-angled, V-shaped wear facet (Chatterjee & Zheng, 2002). Among the teeth bearing paired, V-shaped wear facets, IVPP V17768.20 and IVPP V8299.3 exhibit markedly asymmetrical facets, the distal edge being more deeply worn and developing a pronounced shoulder (Figs. 19A, 19B and 20GG–20HH). That most tooth wear consists of low-angled, V-shaped wear facets suggests an interlocking upper and lower jaw occlusion typical of eusauropods (Chatterjee & Zheng, 2002; Carballido et al., 2017). In IVPP V17768.20, the enamel has been worn away basally, and the paired wear facets meet on the lingual aspect of the tooth. In no tooth do the wear facets extend appreciably onto the labial surface.

The left dentary preserves three teeth, though the middle tooth is poorly preserved and morphologically uninformative (Fig. 17E). Only the upper part of the crowns of the dentary teeth are exposed lingually, and no wear facets are apparent on these. Like the maxillary and isolated teeth, the dentary teeth exhibit fine wrinkling of the enamel, bear a low apicobasal ridge that extends to the tip of the tooth and divides the lingual surface in two, and are gently recurved distally. However, the preserved dentary teeth differ from the maxillary and isolated teeth in at least two regards. First, though somewhat transversely crushed, the dentary teeth appear to lack the lingual excavation of the crown that gives the maxillary teeth their spoon-shaped morphology as well as the paired troughs on either side of the midline apicobasal ridge, and the apical portions of their crowns are overall slightly convex lingually. Second, the dentary teeth exhibit marginal denticles, which are entirely absent in all preserved maxillary and isolated teeth. Both well-preserved dentary teeth preserve five enamel tuberosities along their mesial edges, and two to three small tuberosities (including the apical denticle) along their distal edges, as in O. tianfuensis (He, Li & Cai, 1988: Figs. 16 and 17) and Abrosaurus (Ouyang, 1989; ZDM 5033).

Denticles are plesiomorphic for Sauropodomorpha and are generally reduced towards Neosauropoda (Upchurch, Barrett & Dodson, 2004), though they are homoplastically present in some brachiosaurids (Janensch, 1935–1936; Marpmann et al., 2014; Mannion, Allain & Moine, 2017) and may develop plastically within species or individuals, as in Giraffatitan (Janensch, 1935–1936) and Abrosaurus (Ouyang, 1989). That all erupting or replacement maxillary teeth in Bellusaurus lack denticles suggests that denticles may be unique to replacement teeth in the middle of the dentary tooth row, or perhaps more generally to teeth of the lower jaw. Such distinct upper and lower jaw tooth morphologies are not uncommon in sauropods. Denticles are most strongly, but not exclusively, developed in the dentary teeth in Abrosaurus (Ouyang, 1989), and some sauropods have notably larger upper jaw than lower jaw crown diameters (Ouyang & Ye, 2002; Wilson, 2005; Whitlock, Wilson & Lamanna, 2010; Mannion et al., 2013) or show differences in crown cross-sectional shape and curvature (Wilson, 2005) or wear facet pattern and orientation between upper and lower dentitions (Martínez et al., 2016).

Discussion

The abundance of cranial and postcranial material now assignable to Bellusaurus makes the taxon among the more completely known sauropods. However, interpretation of this material for phylogenetic and morphofunctional analyses is potentially complicated by the dissociation of almost all elements and by the juvenile status of the material. As pointed out above, all duplicate elements within the bone bed are of subequal size (Table 1), and among long bones, the smallest exemplars of an element are at least 77% the length of the largest element, and typically more than 85% of this length (Andrew J. Moore, 2017, unpublished data). In concert with histological information indicating that individuals in the bone bed were within two years of hatching at time of death (Mo, 2013), these data suggest that inter-element ratios (e.g., limb proportions; skull proportions) of average linear measurements are likely to be reasonably accurate reflections of the body proportions of young juvenile Bellusaurus. In the absence of adult specimens, what is less clear is how to parse features of the known Bellusaurus material that are likely to reflect adult morphologies from those that might be expected to show substantial ontogenetic variation based on patterns in other sauropod taxa (Table 4).

Table 4 Sauropod taxa for which unambiguously juvenile cranial material is known.

Taxon	Material	Key reference	
Non-neosauropods	
Patagosaurus*	Mandible	Bonaparte (1986)	
Chebsaurus	Premaxilla, basicranium, surangular	Läng & Mahammed (2010)	
Daanosaurus	Maxilla, frontal, parietal, quadrate, otoccipital	Ye, Gao & Jiang (2005)	
Moabosaurus*	Braincase	Britt et al. (2017)	
Macronaria	
Bellusaurus	Maxilla, nasal, frontal, parietal, squamosal, quadrate, pterygoid, otoccipital, basisphenoid, dentary, angular, teeth	Dong (1990); this study	
Camarasaurus*	Complete skull; isolated premaxilla	Gilmore (1925) and Britt & Naylor (1994)	
Europasaurus*	Nearly complete skull elements from numerous individuals	Marpmann et al. (2014)	
Auca Mahuevo titanosaurian	Largely complete embryonic skulls	Salgado, Coria & Chiappe (2005), García et al. (2010) and García & Cerda (2010)	
Lithostrotia indet.	Partial embryonic skull	Grellet-Tinner et al. (2011)	
Rapetosaurus*	Quadrate, pterygoid, partial braincase, surangular, angular	Curry Rogers & Forster (2004)	
Vahiny*	Basioccipital	Curry Rogers & Wilson (2014)	
Saltasaurus*	Frontal	Powell (1992)	
“Astrodon”	Maxilla, fragmentary pterygoid partial braincase elements, dentary, teeth	Carpenter & Tidwell (2005) and D’Emic (2013)	
Diplodocoidea	
cf. Diplodocus*	Largely complete skull	Whitlock, Wilson & Lamanna (2010)	
Note:

Taxa for which adult cranial material is also known are indicated with an asterisk.

Until recently, much of what was known about sauropod skull anatomy relied on a handful of exceptional specimens (Marsh, 1884; Holland, 1924; Gilmore, 1925; Janensch, 1935–1936; Nowinski, 1971; Berman & McIntish, 1978; Chatterjee & Zheng, 2002). A spate of recent discoveries and redescriptions has dramatically expanded our understanding of sauropod cranial diversity and morphology (Chure et al., 2010; Whitlock, Wilson & Lamanna, 2010; Knoll et al., 2012; Royo-Torres & Upchurch, 2012; Poropat & Kear, 2013; Tschopp & Mateus, 2013; Xing et al., 2015a; Marpmann et al., 2014; Sues et al., 2015; Martínez et al., 2016; Wilson et al., 2016; Britt et al., 2017; Tschopp & Mateus, 2017), but ontogenetic variants of sauropod skulls remain exceedingly rare, and relatively little is known about how the sauropod skull changed as it developed.

Below, we review the state of knowledge of ontogenetic trends in sauropod skulls, summarizing major hypothesized transformations in Table 5, and discuss the implications of the juvenile status of the available Bellusaurus material for interpretation of several cranial characters that likely inform its phylogenetic position.

Table 5 Hypothesized transformations in the course of cranial ontogeny in sauropods.

Hypothesized transformation	Taxa providing evidence	Sources	Comments	
Extreme decrease in relative skull size	Sauropodomorpha	Ikejiri (2004), Rauhut et al. (2011), this study	While relative skull size decreases through ontogeny in many vertebrates, the exceptional diminution of skull size on the evolutionary line leading to Eusauropoda (Rauhut et al., 2011) indicates that negative allometry of the skull with respect to the postcranial skeleton in sauropods was especially extreme	
Increasingly fused interfrontal and interparietal sutures	Diplodocoidea	Salgado (1999), Whitlock, Wilson & Lamanna (2010), Tschopp & Mateus (2013) and Tschopp, Mateus & Benson (2015)	–	
Decrease in relative anteroposterior width of nasal process of premaxilla	Camarasaurus; Europasaurus	Britt & Naylor (1994) and Marpmann et al. (2014)	–	
Stepped margin of the muzzle becomes more distinct	Camarasaurus; Shunosaurus	Britt & Naylor (1994) and Wilson & Sereno (1998)	This hypothesized transformation does not apply to taxa lacking a stepped muzzle (e.g., diplodocids)	
Muzzle increases in relative width	cf. Diplodocus; Camarasaurus; Europasaurus	Whitlock, Wilson & Lamanna (2010), Whitlock (2011b) and Marpmann et al. (2014)	–	
Muzzle transitions from rounded and narrow to square and blunt	Diplodocinae	Whitlock, Wilson & Lamanna (2010) and Tschopp & Mateus (2013)	In addition to general widening of the muzzle, the anterior, tooth-bearing region of diplodocids transitions from transversely narrow and rounded to square and blunt. This transition has been suggested to signify niche partitioning between juveniles and adults, and is thus far hypothesized only for diplodocids, or possibly Diplodocoidea generally	
Elongation of the snout	Theropoda; Massospondylus; cf. Diplodocus; Titanosauria	Varricchio (1997), Salgado, Coria & Chiappe (2005), Whitlock, Wilson & Lamanna (2010), García et al. (2010) and K. E. Chapelle, 2016, unpublished data	Such a transformation is typical of theropods and basal sauropodomorphs. However, rostral elongation was diminished in eusauropods, followed by subsequent elongation of the rostrum in diplodocoids and some advanced titanosauriforms (see Discussion)	
Posterodorsal retraction of the external nares	Titanosauria	Salgado, Coria & Chiappe (2005) and García & Cerda (2010)	–	
Reduction of the maxillary process of the jugal	Titanosauria	Salgado, Coria & Chiappe (2005)	–	
Decreasing participation of jugal in ventral margin of the skull	Europasaurus; Titanosauria	Marpmann et al. (2014), Salgado, Coria & Chiappe (2005) and García et al. (2010)	The presence of a free ventral edge of the jugal in adult Europasaurus, though reduced from the juvenile condition, is hypothesized to reflect paedomorphosis. The jugal has a large participation in the ventral margin of the skull in embryonic titanosaurians, a character otherwise absent in sauropods more derived than Shunosaurus, with the exception of Europasaurus, Giraffatitan, and Malawisaurus (Royo-Torres & Upchurch, 2012)	
Frontal becomes relatively wider and less elongate	Diplodocidae; Europasaurus; Titanosauria	García et al. (2010), Tschopp & Mateus (2013), this study	The observation that the frontal is relatively longer anteroposteriorly in the paedomorphic Europasaurus, embryonic titanosaurians, and some juvenile sauropods (Bellusaurus, Daanosaurus) than is typical for sauropods suggests that this feature may vary ontogenetically	
Decrease in depth of orbital rim penetration of the frontal (in dorsal view), perhaps with concomitant increase in width of frontal, including articulations with prefrontal and nasal	Europasaurus; Bellusaurus; Daanosaurus; Titanosauria; possibly cf. Diplodocus	Marpmann et al. (2014), Chiappe et al. (1998), García et al. (2010), Whitlock, Wilson & Lamanna (2010), this study	See Discussion	
Decrease in distance separating the supratemporal fenestrae	Europasaurus; Camarasaurus	Marpmann et al. (2014), this study	–	
Decrease in relative size of supratemporal fenestrae	Titanosauria	Salgado, Coria & Chiappe (2005)	The supratemporal fenestra is well-developed in titanosaurian embryos but is reduced in more adult titanosaurian skulls, suggesting ontogenetic reduction of this aperture in titanosaurians	
Opening or closure of frontal/frontoparietal/parietal fenestra?	Camarasaurus; Europasaurus; Apatosaurus; Titanosauria	Salgado, Coria & Chiappe (2005), Balanoff, Bever & Ikejiri (2010), Marpmann et al. (2014) and Woodruff & Foster (2017)	See Discussion	
Increasingly large separation between squamosal and quadratojugal	Flagellicaudata	Whitlock, Wilson & Lamanna (2010) and Tschopp & Mateus (2013)	Flagellicaudatans lack contact between the squamosal and quadratojugal, while these two elements are nearly touching in a juvenile diplodocid (CM 11255), suggesting this feature may vary ontogenetically. Notably, embryonic titanosaurians lack the squamosal-quadratojugal articulation present in more adult titanosaurians (Salgado, Coria & Chiappe, 2005), possibly indicating lineage-specific ontogenetic trajectories	
Increasing depth of quadrate fossa	Europasaurus	Marpmann et al. (2014)	This mirrors a phylogenetic increase in the depth of this fossa in Sauropodomorpha (reversed in Flagellicaudata)	
Increasing development of tooth-like process on body of pterygoid	Europasaurus	Marpmann et al. (2014)	–	
Increasing depth of basipterygoid fossa of the pterygoid	Europasaurus	Marpmann et al. (2014)	–	
Crista interfenestralis ossifies	Apatosaurus; Nebulasaurus; Bellusaurus	Balanoff, Bever & Ikejiri (2010), Xing et al. (2015a), this study	–	
Ossification of bony septum dividing the optic (CN II) foramen into separate foramina	Europasaurus; basal sauropodomorphs	Marpmann et al. (2014)	The presence of a single optic foramen in basal sauropodomorphs and apparent reversion to the plesiomorphic state in Europasaurus suggests ontogenetic ossification of this septum	
Closure of external mandibular fenestra	Titanosauria	Salgado, Coria & Chiappe (2005) and García et al. (2010)	An external mandibular fenestra is absent in neosauropods, with the exception of Nigersaurus (Sereno et al., 1999, 2007). Appearance of this feature in embryonic titanosaurians suggests it is lost over the course of ontogeny, though without adult representatives of the Auca Mahuevo titanosaurian, this remains speculative	
Decrease in relative size of surangular foramen	Rapetosaurus	Curry Rogers & Forster (2004)	–	
Displacement anteriorly of the posterior extent of the maxillary tooth row	Diplodocinae; Titanosauria	Salgado, Coria & Chiappe (2005), García & Cerda (2010), García et al. (2010), Whitlock, Wilson & Lamanna (2010) and Tschopp & Mateus (2013)	–	
Decrease in number of teeth?	Camarasaurus; Bonitasaura; Diplodocidae	This study	See Discussion	
Note:

This list is not exhaustive, and emphasizes ontogenetic hypotheses that are particular to sauropods and can be made for more than one taxon.

Sauropod cranial ontogeny

Homology and ontogeny of the preantorbital opening

Understanding patterns of ontogenetic variation and evolutionary elaboration of the preantorbital opening requires insight on the morphogenetic antecedents of this structure. Before discussing evidence for ontogenetic variability in the preantorbital opening, we first review new information on the primary homology (sensu De Pinna, 1991) of the foramen and summarize criteria for its identification.

Recent studies have argued that the definitive preantorbital fenestra of diplodocids is vascular in origin. This work posits that the broad window into the antorbital cavity that is present in various advanced neosauropods arose from a neurovascular opening on the lateral surface of the maxilla, and provides topological criteria for establishing the primary homology of these openings, which, as discussed above, can be quite disparate in form (W. R. Porter, 2015, unpublished data; Martínez et al., 2016). These homology criteria are: (1) the foramen/fenestra is located dorsal to the maxillary palatal shelf, where it communicates with the canal for the maxillary neurovascular bundle; (2) the foramen/fenestra is in the vicinity of the suborbital fenestra, (i.e., where the palatine and ectopterygoid unite with the maxillary palatal shelf); and (3) the foramen/fenestra is generally just posterior to the alveolar tooth chamber (which houses the replacement teeth and may extend somewhat posterior to the most distal [= posterior] erupted tooth position) (Martínez et al., 2016).

Previous study of Bellusaurus overlooked the presence of a large neurovascular foramen anteroventral to the antorbital fenestra (Mo, 2013), which we recognize as the preantorbital foramen. We note that the preantorbital foramen in Bellusaurus does not meet all of the homology criteria proposed by Martínez et al. (2016): while the foramen communicates with the canal for the maxillary neurovascular bundle (and hence with both the anterior maxillary foramen and posterior alveolar foramen), it is at the level of, rather than fully dorsal to, the maxillary palatal shelf, and is well anterior of both the suborbital fenestra and the posterior margin of the alveolar tooth chamber. However, some of the taxa that Martínez et al. (2016) highlight as having a preantorbital foramen also appear to violate these criteria. In Giraffatitan (MB.R.2180.2) and Abydosaurus (DINO 17849), the preantorbital fenestra is ventral or level with the maxillary palatal shelf, while in Euhelopus, the possible perforation of the maxilla posteroventral to the lacrimal process that has been interpreted as the preantorbital opening (Wilson & Upchurch, 2009; Martínez et al., 2016) lacks an apparent connection to the neurovascular bundle of the maxilla (based on study of CT scans of PMU 24705/1a-b). In addition, the preantorbital foramen is anterior to the posterior extent of the tooth row, and presumably the alveolar tooth chamber, in Giraffatitan (MB.R.2180.2), embryos of the Auca Mahuevo titanosaurian (García et al., 2010), and Abydosaurus (DINO 17849), and is level with, rather than posterior to, the posterior margin of the tooth row in Rapetosaurus (Curry Rogers & Forster, 2004) and perhaps Sarmientosaurus (Martínez et al., 2016: Figs. 6 and 7). With the exception of Euhelopus, we do not question the presence of a preantorbital foramen in these taxa, but rather suggest that the criteria proposed by Martínez et al. (2016), may be overly stringent, and principally reflect the topological relationships of the preantorbital opening in adult specimens of diplodocids (e.g., Galeamopus) and titanosaurians (e.g., Tapuiasaurus) for which the identity of this structure is unambiguous. Our current state of knowledge is insufficient to assess whether specific topological information for the preantorbital opening is static through development, though conjectural hypotheses for ontogenetic transformations of the maxilla of titanosaurians (García & Cerda, 2010: Fig. 7) suggest substantial ontogenetic variability is plausible. In asserting the primary homology of the large neurovascular foramen on the lateral surface of the maxilla of Bellusaurus with the preantorbital opening that has been recovered as a synapomorphy of Neosauropoda + Jobaria (Wilson, 2002), we adopt part of the first criterion of Martínez et al. (2016)—that is, connectivity with the maxillary neurovascular bundle—and also draw on criteria that have been used by previous authors (Wilson & Sereno, 1998; Curry Rogers & Forster, 2004) and that obtain for all taxa listed by Martínez et al. (2016), with the exception of Euhelopus—specifically, the relatively large size of the opening vis-à-vis other maxillary neurovascular foramina, the general positioning of the opening anteroventral to the antorbital fenestra, and the anteroventral orientation of the associated channel.

Little is known about the ontogeny of the preantorbital opening. Wilson & Sereno (1998) noted that the slit-like preantorbital opening of Camarasaurus is sometimes reduced in size or essentially absent in adult specimens, but an exhaustive sampling of this feature across Camarasaurus specimens has not yet been conducted, and the generally weak development of this neurovascular foramen even in those specimens that clearly have it (e.g., CM 11338; CM 113; DMNH 32162, a cast of GMNH [Gunma Museum of Natural History] PV 101) make putative ontogenetic transformations difficult to assess. The presence of a distinct preantorbital opening in one maxilla of Bellusaurus (IVPP V17768.3) but not another (IVPP V17768.1) could suggest slight individual variation in the timing of the development of the foramen. Alternatively, this difference could reflect population-level variation in the presence of the opening, or, depending on the phylogenetic affinities of Bellusaurus, the presence of a protracted “zone of variability” (Bever, Gauthier & Wagner, 2011) near the base of Neosauropoda resulting in sustained polymorphism across several nodes; the latter explanation could also account for the inconsistent development of the opening in Camarasaurus specimens. The development of the preantorbital fenestra in embryonic titanosaurians (García, 2007) and its apparent persistence as a well-developed aperture in more adult titanosaurians suggests that this feature arose early in titanosaurian ontogeny and persisted through development, though this remains conjectural until adult individuals of the Auca Mahuevo titanosaurian are discovered.

Ontogeny of the tooth row and snout

Previous studies of diplodicines and titanosaurians indicate that the posterior extent of the maxillary tooth row becomes displaced anteriorly over the course of ontogeny in these long-snouted lineages (Salgado, Coria & Chiappe, 2005; García & Cerda, 2010; García et al., 2010; Whitlock, Wilson & Lamanna, 2010; Tschopp & Mateus, 2013; Table 5). The disparity in the extent of the tooth row and the form of the maxilla that has been hypothesized to exist between embryonic and adult titanosaurians is extreme (García & Cerda, 2010: Fig. 7), and seems likely to predict ecological separation of young juveniles and adults. The muzzle was noted to increase in relative width through ontogeny in Europasaurus (Marpmann et al., 2014), and in cf. Diplodocus, differences in muzzle shape between juvenile and more adult specimens have been suggested to reflect ontogenetic resource partitioning (Whitlock, Wilson & Lamanna, 2010), a pattern that has also been proposed as a possible explanation for divergent patterns of enamel microwear between juvenile and adult Camarasaurus teeth (Fiorillo, 1998).

Relevance to interpretations of Bellusaurus. It is possible that the posterior extent of the maxillary tooth row migrates somewhat farther anteriorly in more adult specimens of Bellusaurus, especially if adults developed a more elongate rostrum than the shape inferred for juvenile Bellusaurus (Fig. 1). The morphologically similar taxon Camarasaurus shows little or no tooth row migration through ontogeny, though the youngest relevant Camarasaurus material (CM 11338) may not reflect the condition present at the beginning of post-hatching ontogeny, a caveat that likely hinders other inferences of cranial ontogeny in this taxon. Nevertheless, we predict that the posterior margin of the maxillary tooth row remains posterior to the anterior edge of the antorbital fenestra in adult specimens of Bellusaurus, as in eusauropods other than diplodocoids, some brachiosaurids, and some titanosaurians.

Ontogeny of the frontal

The presence of an elongate frontal in basal sauropodomorphs (e.g., Wilson, 2002: character 20) and juvenile and paedomorphic sauropods (Table 5) but not in more adult sauropod specimens suggests that the frontal becomes relatively broader transversely through ontogeny. Marpmann et al. (2014) include in the diagnosis of Europasaurus an anteroposteriorly long frontal with a very deep orbital rim causing an extreme reduction of the frontal-prefrontal and frontal-nasal articulations; as noted above, however, this feature is not unique to Europasaurus, and occurs in a variety of other sauropodomorphs (Table 2). Several observations suggest that the form of the lateral margin of the frontal may change through ontogeny in some sauropods. The juvenile Camarasaurus CM 11338 and several other Camarasaurus specimens (Madsen, McIntish & Berman, 1995: Figs. 14A and 14B; McIntosh et al., 1996: Fig. 10; Woodruff & Foster, 2017: Fig. 4) have a concave lateral margin of the frontal, while this margin is weakly concave or entirely straight in some large braincases referred to Camarasaurus (UMNH VP 5668; Madsen, McIntish & Berman, 1995: Figs. 25A–25C). Moreover, the observation that the frontal is more deeply embayed in basal sauropodomorphs, the paedomorphic Europasaurus, and the juvenile Bellusaurus and Daanosaurus than in most sauropods (Table 2), as well as the presence of an embayed orbital margin in embryonic but not adult titanosaurians (Chiappe et al., 1998; Wilson, 2005; García et al., 2010) and a relatively less well-developed lateral margin of the frontal in a juvenile cf. Diplodocus (Whitlock, Wilson & Lamanna, 2010), suggests that curvature of the orbital margin may be ontogenetically variable. We note, however, that the expression of a concave lateral margin in at least some relatively adult mamenchisaurid and diplodocoid skulls (Table 2) indicates that this feature may vary both ontogenetically and phylogenetically (see below).

Relevance to interpretations of Bellusaurus. We hypothesize that the frontal becomes relatively less elongate and broader transversely in more adult specimens of Bellusaurus. It is not currently possible to predict whether the orbital rim of the frontal would become less concave in adult Bellusaurus.

Ontogeny of the supratemporal fenestra

In Camarasaurus, the breadth of the skull roof separating the supratemporal fenestrae appears to diminish through ontogeny as the fenestrae expand in relative transverse width, a trend also noted for Europasaurus (Marpmann et al., 2014). In the juvenile Camarasaurus (CM 11338), the supratemporal fenestrae are separated by over twice their longest diameter, while in at least some more adult skulls (e.g., USNM 13786; UMNH VP 5669; Woodruff & Foster, 2017), these widths are more nearly equal.

Relevance to interpretations of Bellusaurus. If, as in Camarasaurus and Europasaurus, the ratio of the distance separating the supratemporal fenestrae to the transverse width of the supratemporal fenestra decreases through ontogeny in Bellusaurus, then the juvenile Bellusaurus individuals described here may have already undergone much of this change, as the ratio in these specimens (∼1.2) is far removed from the value observed in the most juvenile Camarasaurus (∼2.0; CM 11338) and Europasaurus (∼2.5; Marpmann et al., 2014: Fig. 7E) specimens, and is instead more typical of the ratio observed in many non-diplodocoid eusauropods.

Ontogeny and homology of the frontoparietal fenestra

Previous authors have suggested that the presence of a frontoparietal fenestra at the juncture of the frontals and parietals of Apatosaurus (Balanoff, Bever & Ikejiri, 2010) and dicraeosaurids (Salgado & Calvo, 1992) may reflect the paedomorphic retention of the embryonic frontoparietal fontanelle, possibly as a result of anterodorsal expansion of the superior sagittal sinus into the space between the developing dermal roof elements (Balanoff, Bever & Ikejiri, 2010). This hypothesis of homology implies closure of the frontoparietal fenestra over the course of ontogeny for those sauropods lacking such an aperture, and may garner support from the observation that a midline aperture between the paired frontals and parietals is also present in titanosaurian embryos (Salgado, Coria & Chiappe, 2005) but absent in the skulls of more mature titanosaurian specimens of Nemegtosaurus (Wilson, 2005) and possibly Tapuiasaurus (Wilson et al., 2016). Among theropods, a frontoparietal fenestra has been identified in the juvenile compsognathid Scipionyx (Dal Sasso & Maganuco, 2011), embryonic paleognath Aepyornis (Balanoff & Rowe, 2007), and extant juvenile birds, and may be present in the perinate troodontid Byronosaurus (Bever & Norell, 2009), but is otherwise absent in more adult members of these lineages, suggesting an ontogenetic explanation for the aperture. The postparietal foramen—a median aperture between the parietals and supraoccipital that is unique to some sauropods, and distinct from the frontoparietal fenestra—has also been suggested to result from paedomorphic retention of an embryonic fontanelle (Salgado, 1999), but the influence of ontogeny on the development of the postparietal foramen is unclear (Tschopp & Mateus, 2013).

The hypothesis of a fontanelle origin for the frontoparietal fenestra is challenged, however, by morphologies observed in some other sauropods. In Camarasaurus, the “frontoparietal” fenestra may be bounded entirely by the frontals (and is thus termed the “frontal aperture” by Woodruff & Foster (2017)), while a similar median opening in Europasaurus is wholly surrounded by the paired parietals and is termed the “parietal fenestra” (Marpmann et al., 2014), presumably implying primary homology with the parietal foramen associated with the photoreceptive parietal (pineal) organ in various extinct and extant vertebrates (Janensch, 1935–1936; Edinger, 1955). Notably, the definitive parietal foramen of some squamates may be present between the paired frontals, between the paired parietals, or at the frontoparietal juncture, and these placements can be intraspecifically variable (Edinger, 1955). The absence of a midline aperture in the dermal skull roof of the most juvenile Camarasaurus skull known (CM 11338) and the sporadic presence of such an opening in various skulls referred to Camarasaurus (White, 1958; Madsen, McIntish & Berman, 1995; Woodruff & Foster, 2017) allows for the possibility that the fenestra opens, rather than closes, over the course of ontogeny (Woodruff & Foster, 2017), at least in some sauropods. A frontoparietal fenestra is also absent in Daanosaurus (ZDM 0193), a Chinese sauropod of unclear affinities that is known only from a partial juvenile skeleton subequal in size to Bellusaurus. Thus, at present, it is not possible to make definitive statements about the ontogeny—or even the homology—of the frontal/frontoparietal/parietal fenestrae of sauropods, though exhaustive study of skulls referred to Camarasaurus, discovery of additional juvenile sauropod skulls, and greater focus on unambiguous osteological correlates of the skull roof aperture for the photoreceptive pineal body may help clarify these issues (Woodruff & Foster, 2017). We note that, as pointed out by Harris (2006), while a parietal fenestra allows for light to reach the pineal body, the imposition of a large dural venous system between the brain tissue and dorsal skull roof would presumably hinder photoreception, unless the pineal body was distinct anteriorly from more posterior expansions of the superior sagittal sinus (Balanoff, Bever & Ikejiri, 2010).

Relevance to interpretations of Bellusaurus. Given the ambiguity surrounding the ontogeny and homology of the preantorbital and postparietal fenestrae in sauropods, we refrain from making predictions about the fate of these apertures in more adult specimens of Bellusaurus.

Ontogeny of frontal-parietal fusion

The frontal and parietal appear to be among the first bones to become tightly sutured within the sauropod skull. This has previously been suggested for Camarasaurus (Madsen, McIntish & Berman, 1995) and is evidenced by the strong suturing in the three frontal-parietal pairs of Bellusaurus described above and the two frontal-parietal pairs known for the juvenile Chinese sauropod Daanosaurus (ZDM 0193). That the four frontals and five parietals known for Europasaurus were found in isolation is thus especially striking, and the general lack of inter-element fusion in Europasaurus, even between apparently adult elements, is likely an effect of cranial paedomorphosis (Marpmann et al., 2014).

Relevance to interpretations of Bellusaurus. Assuming a subequal age for the cranial material of Bellusaurus, we hypothesize that strong suturing of the frontal and parietal preceded strong suturing and fusion of the parabasisphenoid to adjacent elements.

Ontogeny of the quadrate fossa

Among the element-specific ontogenetic transformations detailed by Marpmann et al. (2014) for Europasaurus is increasing depth of the quadrate fossa in more mature specimens (Table 5). There is little difference in the depth of the quadrate fossa between juvenile and adult Camarasaurus, though this may reflect the completion of an ontogenetically progressing embayment of the quadrate in juvenile Camarasaurus CM11338 rather than the lack of an ontogenetic transformation in the taxon.

Relevance to interpretations of Bellusaurus. The quadrate fossa is deeply excavated in Bellusaurus and likely did not change much from the condition in the specimen described here, suggesting that ontogenetic embayment of the quadrate was accomplished by the first year or so of life.

Ontogeny of the crista interfenestralis

As noted above, the broad confluence between the metotic and vestibular fenestrae in Apatosaurus (Balanoff, Bever & Ikejiri, 2010) and Nebulasaurus (Xing et al., 2015a) may result from late postnatal ossification of the crista interfenestralis (Balanoff, Bever & Ikejiri, 2010).

Relevance to interpretations of Bellusaurus. In Bellusaurus, separation of the metotic foramen and foramen vestibuli is nearly complete, with only a small ventral gap uniting these apertures. If incomplete separation of the metotic foramen and foramen vestibuli indeed results from late postnatal ossification of the crista interfenestralis, we would predict the closure of this gap to occur in more adult specimens of Bellusaurus.

Ontogeny of tooth count

Ikejiri, Tidwell & Trexler (2005) demonstrate that putative Camarasaurus lentus specimens show some variation in number of alveoli of the maxilla and dentary, and conclude that these differences are “due to individual variation because the juvenile (CM 11338) has more aveolae [sic] than the large WDC [Wyoming Dinosaur Center] specimens.” McIntosh et al. (1996) also attributed tooth count variation in Camarasaurus to individual rather than ontogenetic variation. Maxillary tooth count (eight to nine) is consistent between the juvenile CM 11338 and the adult WDC Camarasaurus described by Ikejiri, Tidwell & Trexler (2005), but CM 11338 has more dentary teeth (13) than does the WDC Camarasaurus (11 on the left, 10 on the right). A possibly sub-adult specimen of Kaatedocus has more maxillary (≥12) and dentary teeth (12–13) than is typical for more adult diplodocid specimens (Tschopp & Mateus, 2013), which have 10–11 teeth each in the maxilla and dentary (Whitlock, Wilson & Lamanna, 2010). Citing apparent constancy in tooth count between a juvenile and more adult cf. Diplodocus skulls (Whitlock, Wilson & Lamanna, 2010), Tschopp & Mateus (2013) considered the larger tooth counts in Kaatedocus to be due to taxonomic rather than ontogenetic variation; however, preservation of the tooth rows in the juvenile diplodocid described by Whitlock, Wilson & Lamanna (2010) is imperfect and tooth counts for this individual were estimated based on comparison to more adult skulls, suggesting the possibility of unrecognized ontogenetic variation in tooth count.

While more data are needed to robustly test the hypothesis that Camarasaurus, diplodocids, or other sauropods lost teeth through development, such a scenario could be consistent with the phenomenon of ontogenetic tooth loss present in many disparate lineages of vertebrates (Wang et al., 2017b). Notably, the pattern identified by Wang et al. (2017b) implies that lineages that show phylogenetic tooth reduction and include taxa with ontogenetic niche partitioning are expected to exhibit tooth reduction through ontogeny at some point in their evolutionary history. Previous research has demonstrated phylogenetic tooth reduction in Sauropodomorpha (Wilson & Sereno, 1998), and, as discussed above, has suggested ontogenetic resource partitioning for Diplodocus and Camarasaurus. Of particular interest to hypotheses of tooth loss in sauropods is the recently discovered titanosaurian Bonitasaura, the dentary of which appears to have fewer alveoli than other titanosaurians and bears a highly vascularized, edentulous, possibly keratin-covered shearing edge in its posterior half (Apesteguía, 2004; Gallina & Apesteguía, 2011). Nigersaurus also exhibits rostralization of the tooth row and an acuminate, edentulous margin of the dentary, which Sereno et al. (2007) suggest may have been associated with a keratinous sheath. Given that odontogenesis may be inhibited by the antagonistic influence of a keratinized oral appendage in taxa showing ontogenetic tooth reduction, we predict that the development of an edentulous shearing edge of the jaw in Bonitasaura and possibly Nigersaurus entailed progressive loss and/or anterior displacement of posterior alveoli through ontogeny (Wang et al., 2017b), hypotheses that can be tested by future discoveries of juvenile individuals.

Ontogeny of endocranial soft tissues

We are aware of no explicit statements for sauropods concerning the ontogeny of the brain or surrounding tissues (as approximated by physical or digital endocasts), which is unsurprising, given the paucity of well-preserved juvenile sauropod cranial material generally. That the system of large dural venous sinuses typical of most sauropods was well-developed in Bellusaurus and a juvenile Camarasaurus (CM 11338; Sereno et al., 2007; Witmer et al., 2008) indicates that elaboration of these sinuses occurred by relatively early stages of post-hatching development. The transverse ridge on the ventral surface of the parietal that divides the dural venous sinuses into anterior and posterior portions in Bellusaurus, Camarasaurus, and other early-branching macronarians is also present in adult Massospondylus, where it shows ontogenetic development, being incipiently developed or absent in more juvenile specimens (K. E. Chapelle, 2016, unpublished data: Figs. 2.20A and 2.20B; Chapelle & Choiniere, 2018: Fig. 36). The absence of this partition of the dural venous sinuses in the non-neosauropod eusauropods Spinophorosaurus and Qijianglong may indicate loss of this ridge in non-macronarian sauropods, immaturity of the Spinophorosaurus and Qijianglong specimens, or convergent evolution of a complex endocranial ceiling in Massospondylus and macronarians. Regardless, the apparent loss of the transverse ridge in diplodocoids—or its independent gain in Massospondylus and macronarians—serves to highlight that sauropod endocranial anatomy is an under-leveraged source of phylogenetic information (Balanoff, Bever & Ikejiri, 2010). Ultimately, distinguishing the influences of ontogeny, convergence, and intraspecfic development on the distribution of the transverse ridge of the parietal endocranial surface, and on endocranial morphology more generally, will require the discovery of more sauropodomorph skulls and further study of taxa for which skulls from more than one ontogenetic stage are known (e.g., Camarasaurus; Moabosaurus).

Evolutionary heterochrony in sauropods

Apart from possible variation in tooth count and our observation that the distance separating the supratemporal fenestrae diminished through ontogeny, few changes have been proposed to have occurred during the growth of the skull in Camarasaurus, and its skull has been suggested to have developed essentially isometrically (Ikejiri, Tidwell & Trexler, 2005). In the absence of a more complete ontogenetic series or the application of quantitative approaches like geometric morphometrics to study shape variation, this assessment may be premature, though modest craniofacial remodeling was also suggested for Europasaurus on the basis of an apparent lack of dramatic ontogenetic change in the morphology and morphometry of cranial structures (Marpmann et al., 2014). Nevertheless, the acquisition in non-neosauropod eusauropods (e.g., M. youngi) and basal macronarians (e.g., Camarasaurus) of a comparatively boxy, short-snouted skull with large orbits and small antorbital fenestrae (Rauhut et al., 2011)—features that typically characterize juveniles of non-avian dinosaurs (Carpenter, Hirsch & Horner, 1994; Varricchio, 1997; Salgado, Coria & Chiappe, 2005; Bhullar et al., 2012; Foth, Hedrick & Ezcurra, 2016)—suggests substantial paedomorphosis of some aspects of skull form along the stem leading to Neosauropoda, though a dense sampling of sauropod skulls has not been included in previous quantitative analyses of heterochrony in dinosaurs (Bhullar et al., 2012; Foth, Hedrick & Ezcurra, 2016). This apparent transformation contrasts with that observed in non-avian theropod dinosaurs, where several lineages show evidence of peramorphic trends in the skull (Bhullar et al., 2012; Foth, Hedrick & Ezcurra, 2016).

The origin and drivers of cranial paedomorphosis in sauropodomorph dinosaurs are incompletely understood. Eoraptor was suggested to have a paedomorphic skull (Bhullar et al., 2012), and a recent study found the skull of Massospondylus to be paedomorphic with respect to a hypothetical massopodan ancestor (Foth, Hedrick & Ezcurra, 2016), though Massospondylus nevertheless shows significantly non-isometric growth through ontogeny, with heterochrony resulting in greater growth of bones of the snout than those of the braincase and orbit (K. E. Chapelle, 2016, unpublished data). From a presumably paedomorphic, somewhat Camarasaurus-like ancestor, some diplodocoids and advanced titanosauriforms evolved superficially convergent skull morphologies, broadly characterized by an elongate, horse-like rostrum, gracile mandibles, reduced and inclined adductor chambers, shortened tooth rows, and slender teeth (Upchurch, 1998, 1999; Chure et al., 2010; Button, Barrett & Rayfield, 2017). It is not yet known whether these transitions involved element-specific developmental trajectories like those documented for Massospondylus (K. E. Chapelle, 2016, unpublished data), but these long-faced lineages did not re-invade the functional morphospace of their basal sauropodomorph predecessors (Button, Barrett & Rayfield, 2017). Though it is beyond the scope of this review to summarize macroevolutionary trends in sauropod cranial morphology, we conclude our discussion of cranial ontogeny and evolutionary heterochrony by noting that several transformations of the skull hypothesized to occur during sauropod development—including extreme diminution of the skull relative to the postcranial skeleton, an increase in relative muzzle width, anterior displacement of the posterior end of the maxillary tooth row, and posterodorsal retraction of the nares (Table 5)—are recapitulations of shape changes that occurred in the course of sauropodomorph evolution (Rauhut et al., 2011).

Phylogenetic affinities of Bellusaurus

The phylogenetic position of Bellusaurus has been uncertain since its original, brief description over 20 years ago (Dong, 1990). In pre-cladistic treatments, Bellusaurus was placed within the Brachiosauridae by Dong (1990), and was also suggested to have titanosaurian affinities owing to its procoelous anterior caudal vertebrae (Jacobs et al., 1993). More recently, phylogenetic analyses that include Bellusaurus have recovered the taxon just outside Neosauropoda (Wilson & Upchurch, 2009; Mo, 2013; Royo-Torres & Upchurch, 2012) or near the base of Macronaria (Upchurch, Barrett & Dodson, 2004; Carballido & Sander, 2014). A revised phylogenetic hypothesis of Bellusaurus and other Shishugou Formation sauropods is beyond the scope of this study, and will be addressed elsewhere. Instead, we discuss some issues and cranial characters that are likely to be relevant in future phylogenetic studies that include Bellusaurus or other juvenile sauropod specimens as operational taxonomic units.

Previous studies of dinosaurs, synapsids, and whales indicate that juvenile specimens may be vulnerable to stemward slippage in phylogenetic analyses if they lack apomorphies that are acquired during ontogeny (Kammerer, 2011; Campione et al., 2013; Carballido & Sander, 2014; Choiniere et al., 2013; Tsai & Fordyce, 2014; Currie et al., 2016; Wang et al., 2017a). Notably, this phenomenon may be particularly pervasive in large-bodied lineages, where ontogenetic disparity is especially extreme and adult individuals have greater potential to accumulate “terminal” ontogenetic changes (Tsai & Fordyce, 2014; Wang et al., 2017a). Several of the features hypothesized to vary ontogenetically in sauropods (Table 5) have been included in previous phylogenetic analyses and/or used to diagnose taxa. A deeply embayed orbital margin of the frontal, for example, may be a local autapomorphy of Europasaurus (Marpmann et al., 2014), but our survey suggests that the form of the lateral margin of the frontal can be ontogenetically variable, with more juvenile individuals showing a more concave margin (see above). However, the observation that a concave lateral margin of the frontal is homoplastically present in relatively adult skulls of several phylogenetically disparate groups of sauropodomorphs (Table 2; Tschopp, Mateus & Benson, 2015) suggests that scoring juvenile taxa as unknown for this feature may risk erasing important synapomorphic information. Some previous authors (Currie et al., 2016) have cautioned that juvenile specimens should be scored as unknown (“?”) for characters known to vary ontogenetically. We suggest that this approach should be considered just one of a handful of mutually informative approaches to including juvenile taxa in phylogenetic analyses, and that future studies should explore the effectiveness of applying phylogenetic methods that are known to be more robust to homoplasy than equal weights parsimony—specifically, implied weights parsimony (Goloboff, 1993; Goloboff, Torres & Arias, 2017) and Bayesian phylogenetic approaches with appropriate morphological models (Wright & Hillis, 2014; Pyron, 2016; Puttick et al., 2017; but see Goloboff, Torres & Arias, 2017). The merits of implied weighting for accommodating ontogenetically variable characters, which are expected to be relatively homoplasious in matrices that include juvenile taxa, was discussed by Tschopp, Mateus & Benson (2015), who also noted that the recovery of some juvenile sauropod specimens in well-defined clades and with adult specimens suggests that the influence of ontogenetically variable characters may be minimal in sauropods. Implied weighting and Bayesian approaches have the benefit of allowing all character data to be included, and may be especially well-suited to lineages for which information on ontogenetic variation is scant and ontogenetically variable features may not be readily recognizable as such.

Bellusaurus exhibits several cranial characteristics that have been recovered as synapomorphies of Macronaria, Neosauropoda, or a slightly more inclusive group, including: subnarial foramen directed dorsally (Neosauropoda: Upchurch, 1998; Upchurch, Barrett & Dodson, 2004); subnarial foramen within the external narial fossa (Macronaria: Upchurch, 1998); preantorbital opening in the maxilla (Neosauropoda: Wilson & Sereno, 1998, Upchurch, Barrett & Dodson, 2004; Neosauropoda + Jobaria: Wilson, 2002); lateral articulation of the ectopterygoid principally with the maxilla (Neosauropoda: Upchurch, 1998, Wilson & Sereno, 1998, Upchurch, Barrett & Dodson, 2004; Neosauropoda + Jobaria: Wilson, 2002); deep excavation in the posterior surface of the quadrate (Macronaria: Mannion et al., 2013); stepped dorsal margin of the palatine process of the pterygoid (Neosauropoda: Wilson, 2002); fewer than 18 dentary teeth (Macronaria: Wilson, 2002); and loss of denticles on the anterior and posterior margins of the crown (Neosauropoda: Wilson & Sereno, 1998, Wilson, 2002; note that this is only true for the maxillary dentition of Bellusaurus). In addition, we add to this list the partial division of anterior and posterior portions of the dural venous expansion by a transverse ridge of the endocranial ceiling, which we suggest may constitute a synapomorphy of Macronaria that is subsequently diminished in some titanosauriforms (Wilson et al., 2009; Paulina Carabajal, 2012; Knoll et al., 2013; Sues et al., 2015), though additional sampling of sauropodomorph endocranial anatomy and a phylogenetic analysis including this character are necessary to test this hypothesis. Now that radiometric dating has placed Bellusaurus in the earliest Late Jurassic (Fig. 2; see above), phylogenetic analyses recovering Bellusaurus as a basal macronarian imply a less drastic extension of the age of Neosauropoda than was inferred by earlier studies that considered the Shishugou Formation to be as old as Aalenian in age (Upchurch & Barrett, 2005).

Conclusion

The abundance of cranial and postcranial material now assignable to Bellusaurus makes the taxon among the more completely known sauropods. The thorough description of the cranial anatomy of Bellusaurus presented here allows for a hypothesized reconstruction of the skull and provides new data that will facilitate the inclusion of Bellusaurus in analyses of sauropod phylogeny, cranial morphology, and ontogeny. Detailed comparisons to other sauropod taxa confirm that Bellusaurus is not a juvenile specimen of Klamelisaurus or other Middle-Late Jurassic Chinese sauropods, indicate that Bellusaurus is diagnosable by numerous unique autapomorphies from across the skull, and support previous phylogenetic hypotheses that recover Bellusaurus as an early-branching macronarian or close relative of Neosauropoda. We argue that phylogenetic analyses including Bellusaurus or other juvenile specimens should employ implied weights parsimony and Bayesian inference methods, as these approaches are known to be more robust to homoplasy than equal weights parsimony.

Sauropod skulls are rare, and ontogenetic variants of sauropod cranial material are rarer still. Nevertheless, our review of the current state of knowledge of sauropod cranial ontogeny identifies numerous transformations hypothesized to occur in the course of sauropod ontogeny that can be tested by future discoveries of sauropod skulls, and highlights several outstanding questions that can be addressed with the cranial material currently available. In particular, detailed morphometric analysis of the individual skull bones of specimens referred to Camarasaurus, as well as comparative studies of the endocranial anatomy of taxa for which numerous skulls are known (e.g., Camarasaurus, Moabosaurus, diplodocids), will aid in teasing apart the relative contributions of intraspecific variability and ontogeny to cranial variation in sauropods. In addition, comparative histological and anatomical studies of sauropods with extant taxa that exhibit a parietal foramen associated with a photoreceptive pineal organ will aid in elucidating the developmental and functional significance of the midline aperture of the skull roof observed in numerous sauropod taxa, including Bellusaurus.

This paper represents a portion of the doctoral thesis of AJM at the George Washington University. For their hospitality and access to specimens in their care, we wish to thank A. Henrici (CM), J. Choiniere (BPI), B. Britt & R. Scheetz (BYU), J. Sertich (DMNH), A. Millhouse (USNM), R. Royo-Torres (CPT), Zheng F. & Geng BH. (IVPP), M. Belinchón (MCNV), D. Schwarz-Wings & T. Schossleitner (MB.R.), P. Sereno & T. Keillor (University of Chicago), R. Irmis (UMNH), and Jiang S., Peng GZ, and Li F. (ZDM). We are grateful to L. Witmer for providing CT scans of Camarasaurus (CM 11338) and S. Poropat and B. Kear for scans of Euhelopus (PMU 24705). AJM is indebted to Wang S., Qin ZC., Zhou Y., Wang J., Liao JQ., and Li F. for their assistance and unflagging generosity during research trips to Beijing. We thank Feng Y. and Hou YM. for assistance with CT scanning and Wang HJ., Huo YL., Xiang LS., He SC., Li W., Yu T., and Cao RF. for collecting and preparing the fossils. We are grateful to J. Stiegler, K. Poole, D. White, and A. Ruebenstahl for helpful discussions. Thoughtful reviews by Emanuel Tschopp, an anonymous reviewer, and the editor improved an earlier version of this manuscript.

Institutional abbreviations

BP Bernard Price Institute, Johannesburg, South Africa

BYU Brigham Young University Museum of Paleontology, Provo, UT, USA

CM Carnegie Museum of Natural History, Pittsburgh, PA, USA

FCPT-D Fundación Conjunto Paleontológico de Teruel-Dinópolis, Teruel, Spain (plus CPT for the fossil material deposited in the museum [Museo Fundación Conjunto Paleontológico de Teruel])

DFMMh Dinosaurier-Freilichtmuseum Münchehagen/Verein zur Förderung der Niedersächsischen Paläontologie e.V., Rehburg—Loccum, OT Münchehagen, Germany

DINO Dinosaur National Monument

DMNH Denver Museum of Natural History, Denver, USA

IVPP Institute of Vertebrate Paleontology and Paleoanthropology, Beijing, China

MCNV Museo de Ciencias Naturales, Valencia, Spain

MB.R. Museum für Naturkunde, Berlin, Germany

MNBH Musée National Boubou Hama, Niamey, Niger

UMNH Natural History Museum of Utah, Salt Lake City, UT, USA

USNM Smithsonian National Museum of Natural History, Washington, DC, USA

ZDM Zigong Dinosaur Museum, Dashanpu, China.

Additional Information and Declarations

Competing Interests

Author Contributions

Data Availability

The authors declare that they have no competing interests.

Andrew J. Moore conceived and designed the experiments, performed the experiments, analyzed the data, contributed reagents/materials/analysis tools, prepared figures and/or tables, authored or reviewed drafts of the paper, approved the final draft.

Jinyou Mo conceived and designed the experiments, performed the experiments, analyzed the data, contributed reagents/materials/analysis tools, prepared figures and/or tables, authored or reviewed drafts of the paper, approved the final draft.

James M. Clark conceived and designed the experiments, contributed reagents/materials/analysis tools, authored or reviewed drafts of the paper, approved the final draft.

Xing Xu conceived and designed the experiments, contributed reagents/materials/analysis tools, authored or reviewed drafts of the paper, approved the final draft.

The following information was supplied regarding data availability:

CT scan data (TIFF file stacks) for the maxillae (IVPP V17768.1 and V17768.3) and parabasisphenoid (IVPP V8299.2) are reposited on MorphoBank: Project 3122, http://morphobank.org/permalink/?P3122.

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
