# Peer review of "Cranial anatomy of Bellusaurus sui (Dinosauria: Eusauropoda) from the Middle-Late Jurassic Shishugou Formation of northwest China and a review of sauropod cranial ontogeny"

_PeerJ, doi:10.7717/peerj.4881_

## Round 0.1 · original submission · Minor Revisions

The reviewers provide a number of helpful suggestions, which should be addressed point-by-point in review.

For the ratios in Table 2, can you provide the original measurements also? Raw ratios can obscure patterns of allometry, especially if proportions change across ontogeny. The ratios in both Camarasaurus are basically the same, but ratios vary somewhat for other taxa with multiple specimens (e.g., cf. Diplodocus sp. and Bellusaurus), so again it would be good to note if this is just variation of individuals at the same size or potentially size correlated.

Reviewer 1 ·

Basic reporting

I really like to see that paper coming out, as describe a very important, almost unknown, sauropod that has a great relevance for sauropod evolution and possible the origin of some lineages (e.g. Neosauropoda). I found the paper really informative, well written and very well figured.

I found the description extremely interesting and detailed, and I do not have comments on this section, as is well figured as well. Maybe the authors can call figures in some parts of the text (specially to clearly refer to some morphological character), and not solely at the title of the described bone.

In the discussion the authors include a very complete list of ontogenetic changes, based on previous studies and own observations. This section is very informative but probably could be re-accommodate in order to better identify each transformation.
For example, using sub-headings (e.g., in Line 1236 they start describing ontogenetic transformations in the frontal, whereas lines 1248 to 1257 looks more like an introduction to basal knowledgment, and in line 1258 the changes in preantotbital fenestra are described, in line 1272 the tooth number…
Additionally, in line 1511 the authors describe the problems in the identification of the preantorbital foramen. I suggest to move this part of the text to here maybe, as in this part the authors state that this fenestra is present in Bellusaurus, but discussing at the end how to identify it. Also, maybe you can include here the orientation criterion used by Wilson and Serene (posterior maxillary foramen opens posteroventrally whereas the preantorbital fenestra opens anteroventrally).
Taking into account c the similarities observed in the frontal articulation (for nasal and prefrontal) between Bellusaurus and Europasaurus.. Do you think this characterwill be much different in adult Bellusaurus specimen (and therefore still diagnostic of adult Europasaurus) or you beileve that Bellusaurus will not have so strong transformation and that probably this character will be similar to Europasaurus?, questiongin therefore its usage as autapomorphy (depending in the final position of Bellusaurus).


Dividing this in sub-heading will allow clarifying, at the end of each sub-section, its relevance for the interpretations in Bellusaurus (which are now in a different sub-section all the bones together).

In line 1431, as well as in other sections of the manuscript, the authors mention the absence of ontogenetic trends in Camarasaurus, as the case of tooth row migration. My major concern is about the ontogenetic stage of the Camarasaurus skulls, and the possibility of being them too advanced for having great changes, even if they are from “juvenile specimens”. In the axial skeleton of Europasaurus foe example it was observed that most of the final adult characters were present just prior to the “adult” stage, therefore this implies a morphological maturity previous to the neurocentral closure. Could happened something similar in the skull…?

In the phylogenetic section the authors discuses the possible problems in scoring juvenile taxa. In that section, specially lines 1499 to 1510 the authors mention some derived characters observed in the skull of Bellusaurus, and some of them were previously discussed to do not be affected by ontogeny (or affected but, as in the case of the tooth row length, not plausible to change to the plesiomorphic state). I think that this paragraph could be a bit more detailed, mentioning some more detailed the distribution of these derived characters observed in Bellusaurus (e.g., preantorbital fenetrsa present in Jobaria + Neosauropoda).. As I mentioned above the last part of the discussion seems to be more properly placed in the discussion of the preantorbital fenestra, and here solely include its presence in Bellusaurus and its significance in terms of phylogentic position. Here, the authors can also mantion, shortly, the age of Bellusaurus, as one of the most important inferences made with this taxon as a neosauropods, was the Middle Jurassic origin of this clade, whereas the new age here reported does not change so drastically the origin of Neosauropoda (if Bellusaurus finally is a neosauropod).


I really like the tables included. In table 2 maybe you can include some abbreviation making reference to the phylogenetic position and ontogenetic stage of mentioned taxa (for phylogenetic position e.g., BSM, Basal sauropodomorph; BS, Basal Sauropod; ND, Neosauropod diplodocoid; BM, basal Macronarian, TF, Titanosauriform..), and some information on the ontogenetic stage of the specimens, Marpmanet al used MOS, maybe a similar criterion or any other, at least late juvenile, early juvenile, adult…..
In this way will be much more easy for the reader to identify the possible phylogenetic-related changes and ontogenetic-related changes

Experimental design

My single comments are related to the order of the discussion, as mentioned above. The design of the reseacrh, taxa included in the discussion

Validity of the findings

The findings are relevant, and the manuscript will be certainly highly cited.

Additional comments

I hope to see the work published soon.

Good Luck

Jose

·

Basic reporting

This paper is an important contribution to our knowledge of sauropod anatomy, and brings one of the least understood taxa in a crucial point of sauropod evolution (the radiation of the crown group Neosauropoda) onto the stage by describing in great detail its cranial anatomy. Thanks to the detail, this study will be of long-lasting impact in the community and follows or even sets new standards concerning basic reporting. There are a few typos and peculiar formatting decisions (e.g. many specimen numbers are in bold), but nothing serious.

Experimental design

The paper provides original research, clearly states its significance, and puts everything into the context of sauropod anatomy, ontogeny, and phylogeny. It reviews ontogenetically changing characteristics in sauropod skulls in great detail, and adds significant new data.

Validity of the findings

Discussion and stated implications are well-supported by the data. I especially like the part on the influence of ontogenetic characters on phylogeny, and how to best cope with juvenile specimens in a phylogenetic dataset.

Additional comments

Line 127: Bellusaurus, not Bellusaursu
Line 186: you mention that at least three individuals are represented. It is unclear if you intend that these three individuals are within a particular specimen number – further above you mention that there are 17 individuals in total. Please clarify
Line 190 (and many other notions): The study of Carballido & Sander on Europasaurus axial morphology was officially published in print in 2014.
Line 210: I didn’t understand if Bellusaurus has subdivided pleurocoels in the cervicals or Klamelisaurus. If it’s Bellusaurus you might want to state that this is opposite of what would be expected in a juvenile individual (see Carballido et al. 2012; Carballido & Sander 2014 and Tschopp & Mateus 2017 for more detailed information on this).
Line 256: an elongate scapular blade might also be ontogenetic – in the juvenile sauropod SMA 0009, the scapula is very long (Schwarz et al. 2007; Carballido et al. 2012)
Line 273: Bellusaurus really has LAGs in longbones? That’s unusual for what I know
Line 282: Interesting thought that you can distinguish individual from ontogenetic variation. You might want to add a table of the morphological features that change individuals vs. ontogenetically. But maybe this could also be a paper on its own.
Line 284 (and later): catalog numbers of specimens you use for comparisons are often in bold, you should probably “un-bold” them.
Line 302: I can’t really see that lateral expansion of the nasal process of the maxilla you mention here in the figures, it just seems to project posterodorsally from the maxillary body to me
Line 317: the foramen in the nasal process of the maxilla would probably also qualify as local autapomorphy for Bellusaurus if Shunosaurus had the same opening – in the end they’re very far from each other in the phylogeny.
Line 344 (and elsewhere): the specimen number for the cf. Brachisoaurus skull should probably be USNM 5370. Please check and correct throughout the MS. If this is a typo, it might be a good idea to check if there are others too I didn’t spot because I’m not familiar with Asian taxa/specimens.
Line 513: add “a” between “and the medial,” and “less acute facet”
Line 536: the presence of a cartilaginous palpebral element is highly unlikely, because the palpebral and other supraorbital bones are of dermal origin in crocodiles and lizards, so that there is no cartilaginous precursor (see Maidment et al. 2010; Nesbitt et al. 2012). It is true that we proposed this, so feel free to correct it or just delete it (we will address this in a study that is currently in preparation also).
Line 697: delete one of the opening brackets
Line 732: add “the” between “total height of” and “quadrate. In lateral”
Line 891: the exit of the hypoglossal nerve seems to be variable within Camarasaurus: AMNH 5761 and SMA 0002 have a single one, CM 11338 seems to have a double exit (I’m not sure if this is published somewhere, maybe in some study with Larry Witmer that uses CM 11338? Otherwise feel free to cite as pers. comm.)
Line 917: “paroccipital”, not “paroccipitial”
Line 957: “diplodocoids”, not “diplodocids”, because Amargasaurus is a dicraeosaurid. You could also use flagellicaudatan instead.
Line 1012: “Läng”, not “Lang” (in the bibliography it’s correct)
Line 1117 (and elsewhere): the study of Wiersma & Sander was published in print in 2016
Line 1131: delete first closing bracket
Lines 1150-1153: consider using the terminology used by Holwerda et al. (2015), who described for the first time (to my knowledge) in detail tooth wrinkling patterns
Lines 1219-1221: you might want to add also Chure et al. 2010; Whitlock et al. 2010; and Poropat & Kear 2016
Line 1235: “Camarasaurus”, not “Camarsaurus”
Line 1493: Tschopp et al. (2015) mention that the use of implied weighting should downweight ontogenetically variable characters. However, if juveniles are scored for these characters, the potentially misleading information from them still influences the algorithm, so I’m not so convinced anymore that these scores should be included.
Line 1536: “interpreted”, not “intepreted"
Table 2: is the ratio basically a comparison between anterior and posterior width of the frontal?
Table 3: why are many SI values missing even if you have both apicobasal length and transverse width stated?

---

## Round 0.2 · accepted · Accept

Thank you for your close attention to the comments from the reviewers. The manuscript is now suitable for publication, in my opinion.

#